# Synaptic vesicle traffic is supported by transient actin filaments and regulated by PKA and NO

Nicolas Chenouard[1,2], Feng Xuan[1,4] & Richard W. Tsien[1,3✉]

Synaptic vesicles (SVs) can be pooled across multiple synapses, prompting questions about their dynamic allocation for neurotransmission and plasticity. We find that the axonal traffic of recycling vesicles is not supported by ubiquitous microtubule-based motility but relies on actin instead. Vesicles freed from synaptic clusters undergo ~1 µm bouts of active transport, initiated by nearby elongation of actin filaments. Long distance translocation arises when successive bouts of active transport were linked by periods of free diffusion. The availability of SVs for active transport can be promptly increased by protein kinase A, a key player in neuromodulation. Vesicle motion is in turn impeded by shutting off axonal actin polymerization, mediated by nitric oxide-cyclic GMP signaling leading to inhibition of RhoA. These findings provide a potential framework for coordinating post-and pre-synaptic strength, using retrograde regulation of axonal actin dynamics to mobilize and recruit presynaptic SV resources.

[1] NYU Neuroscience Institute and Department of Neuroscience and Physiology, NYU Langone Medical Center, New York, NY 10016, USA. [2] Univ. Bordeaux, CNRS, Interdisciplinary Institute for Neuroscience, IINS, UMR 5297, F-33000 Bordeaux, France. [3] Center for Neural Science, New York University, New York, NY 10003, USA. [4] Present address: Interdepartmental Neuroscience Program, Northwestern University, Evanston, IL 60208, USA. ✉email: Richard. Tsien@nyulangone.org

Presynaptic boutons are identified by a standing population of synaptic vesicles (SVs)[1–3] but vesicles can also be exchanged between nearby synapses of hippocampal neurons[4–12], cortex[13], and frog neuromuscular junction[14]. These trafficking SVs can be synaptically released and recycled, expanding the classical picture of synapse-resident SV pools to include a de-localized SV "superpool"[8,15]. A superpool might support the reallocation of synaptic weights, enabling selective changes in synaptic strength with overall normalization built-in. Full acceptance of the superpool concept has been hampered by a limited understanding of how SVs are moved between synapses and whether such SV exchange is functionally regulated. The long-range traffic of membrane proteins from the endoplasmic reticulum to axon relies extensively on "cytoplasmic transport packets", propelled along microtubules by kinesins and dyneins[16–18]. Whether and how microtubules participate in the trafficking of recycling SVs is less clear[7,12]. Actin-based motility of membranous organelles was established by pioneering work in invertebrate axons[19–23] but transport was envisioned as mostly transverse, to complement microtubule-driven, longitudinal trafficking. Interestingly, in rodent axons, actin forms circumferential rings[24,25] as well as longitudinal filaments up to ~20 µm long that participate in a slow, anterograde actin flow[25,26]. The acto-myosin2 network controls intra-synaptic mobility[11,27] and Myosin V translocation along filamentous (F-)actin may propel SVs along axons[12]. It remains unclear how this reconciles with a prevailing emphasis on microtubule-based vesicular transport or with the abrupt halting of axonal SV transfer by an "actin stabilizer", jasplakinolide[5].

Here we use time-lapse microscopy to demonstrate two distinct mechanisms of vesicle traffic along axons. We confirmed that traffic of transport vesicles rich in the synaptic protein VAMP2/synaptobrevin[16,28,29] was dependent on conventional microtubule-based motility. Surprisingly, actin, but not microtubules, supported the transport of recycling presynaptic vesicles that underwent activity-dependent fusion and retrieval. To characterize the underlying axonal actin structures, we employed specific pharmacological interventions along with imaging of fast, transient actin polymerization[25] and traffic of recycling SVs loaded with single quantum dots (QDs)[9,10,30]. SVs transiently and repeatedly associated with polymerizing actin filaments whose arrival at SVs rate-limited longitudinal SV motion. Our observations provided a unified explanation of previously disparate results using inhibitors of myosin V motor activity[12] and actin depolymerization[5]. Moreover, analysis of single SV dynamics showed that the active transport (AT)-enabled SVs differed from other extrasynaptic SVs in displaying faster free diffusion between AT bouts. Turning to regulatory questions, activation of protein kinase A (PKA), a player in neuromodulation and plasticity[31], increased the initiation of SV AT while leaving actin dynamics intact. In contrast, nitric oxide (NO), an established retrograde messenger[32–36], acutely reduced longitudinal actin polymerization and, in turn, SV traffic, effects mediated by signaling downstream of NO, the elevation of cyclic GMP, and inactivation of RhoA. The same NO $\rightarrow$ cGMP $\rightarrow$ RhoA $\rightarrow$ actin pathway was previously found to augment the clustering of presynaptic vesicular proteins and long-term synaptic potentiation[36]. Thus, our findings provide a mechanistic link between retrograde signaling and presynaptic reallocation of functional resources.

## Results

### Nocodazole spares the traffic of recycling synaptic vesicles. We tracked vesicle mobility by monitoring fluorescent mCherry, fused to VAMP2/synaptobrevin, an integral membrane protein present in both recycling SVs and some "cytoplasmic transport packets"[28,29,37,38]. The construct was delivered by transfection of hippocampal neurons in culture at a stage (DIV 12–16) when functionally mature synapses were evident from staining and destaining of FM dyes, a hallmark of activity-dependent vesicle fusion and recycling[1,2,39–41] (Supplementary Fig. 1). VAMP2: mCherry clustered in puncta of various sizes and intensities (Fig. 1a left), positioned along axons (yellow dashed line). The heterogeneity of VAMP2 + vesicular structures were clarified by prolonged (2–4 h) incubation with concentrated, fluorescently labeled antibodies against a luminal epitope of synaptotagmin 1 (Syt1-IgG:C488), which label recycling SVs[42]. Along with puncta marked by both labels (presumed recycling SVs) (white arrowheads, Supplementary Fig. 2A), we observed puncta strongly labeled by VAMP2:mCherry but lacking detectable Syt1 antibody staining, as expected for "cytoplasmic transport packets" that do not undergo synaptic recycling (Supplementary Fig. 2). Movements of puncta were captured by video recording (3 Hz, 2 min), then presented in kymographs plotting one-dimensional axonal position vs. time (Fig. 1a). Some puncta were approximately stationary (top, blue arrowhead), likely corresponding to synaptic boutons. Others moved intermittently, merged together or split, and ranged widely in size from small (red arrowhead) to large (yellow arrowhead). We compared control images (2 h treatment with 1/1000 DMSO) with images after 2 h treatment with 10 µM nocodazole, which disrupted microtubules in neural processes (verified by α-tubulin immunostaining, Supplementary Fig. 3). Upon microtubule disruption, the motion of mCherry-tagged puncta sharply fell relative to control, seen as largely static puncta (Fig. 1a bottom, yellow and blue), with exceptions (red). Vesicle motility along axons was quantified by computing the autocorrelation of VAMP2:mCherry fluorescence against time and fitting a single exponential for individual movies (Methods). After microtubule disruption (nocodazole, $N = 38$) autocorrelation functions decayed ~4-fold more slowly than in control (DMSO, $N = 22$) (median decay rate $-74.7\%$, $p < 10^{-4}$, Wilcoxon rank-sum (RS) test), reflecting slower changes in fluorescence and reduced motility (Fig. 1b). Relative to control, AT rate of occurrence (number of AT events per unit length of axon per unit observation time) after nocodazole treatment was reduced by 75.5%; velocity and length of individual AT events were reduced by ~45%, while their duration was spared (Supplementary Fig. 5A, B). Overall AT motility (cumulated AT distance divided by observation time and length of monitored axon) of VAMP2:mCherry cargos was sharply decreased by nocodazole (median $-83.6\%$) (Fig. 1c). Interestingly, velocities did not scale uniformly: instead, a high velocity ($\geq 1.5$ µm s$^{-1}$) subpopulation of events disappeared. As the traffic of VAMP2: EGFP transport packets is abolished by nocodazole in the initial segment of immature axons that lack well-formed SVs[16], a logical inference is that nocodazole-resistant VAMP2 clusters correspond to recycling SVs (labeled with Syt1-IgG in Supplementary Fig. 2).

To focus specifically on recycling SVs, neurons were first preincubated with biotinylated Syt1-IgG. To render such vesicles fluorescent, neurons were further stimulated with 45 mM K+ for 90 s in the presence of streptavidin-coated quantum dots (QDs), which are bright, photobleaching-resistant, and taken up 1:1 by SVs, as demonstrated by electron microscopy[30]. Non-specific fluorescence was minimized by external fluorescence extinction with a bath-applied quencher (Supplementary Fig. 6). The improved signal facilitated tracking isolated vesicles as well as larger aggregates of vesicles moving en masse (Fig. 1e). Moreover, the brief exposure to QDs ensured that SVs were still fusion- and recycling-competent, as verified by exocytosis triggered by later electrical or high K+ stimulation (Supplementary Fig. 6)[9,10,30]. In

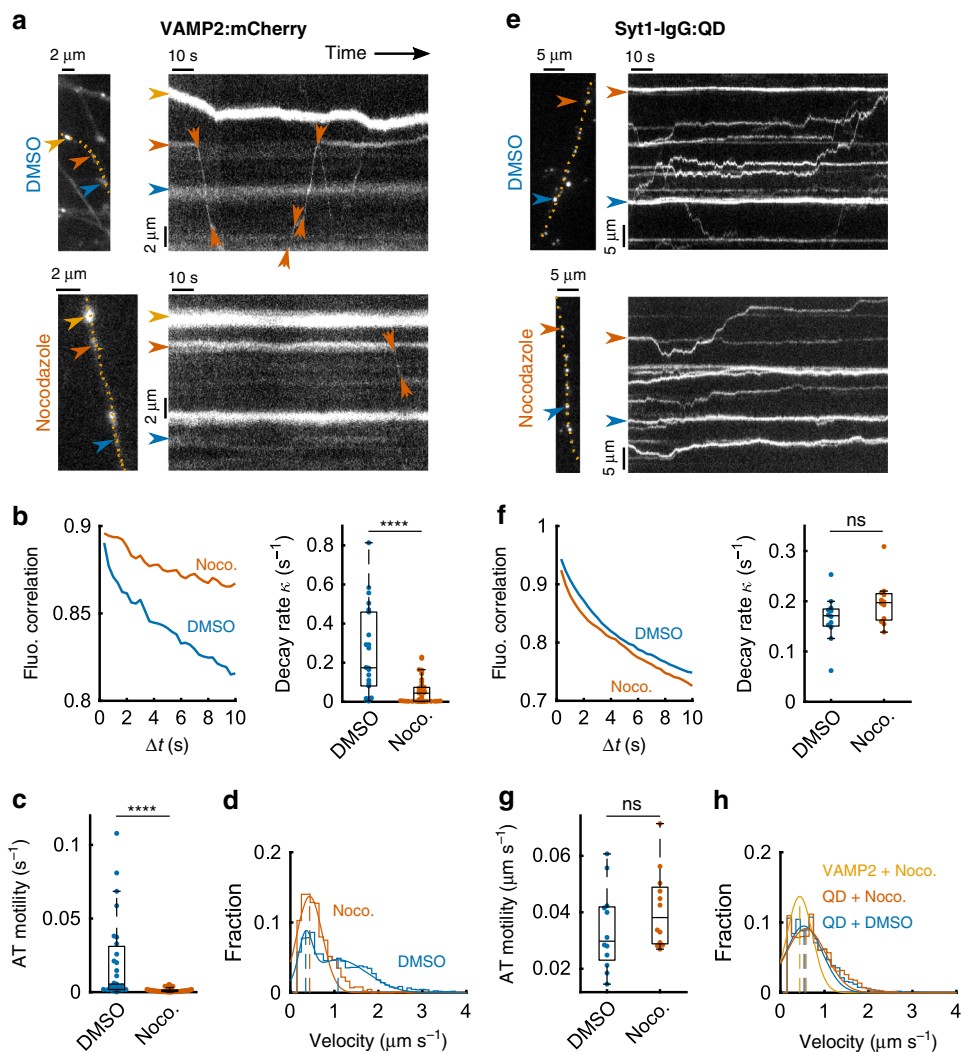

**Fig. 1 Transport of VAMP2:mCherry clusters was impaired by microtubule depolymerization with nocodazole while traffic of SVs was not.** Axonal traffic in hippocampal cultures was monitored by live microscopy after transfection with VAMP2:mCherry or SV labeling with Syt1-IgG:QD. Kymographs represent spatiotemporal variations of fluorescence along with axonal segments (yellow dashed lines). **a** Axons in control conditions (DMSO 1/1000, 2 h) vs treatment with nocodazole (Noco.) (10 μM, 2 h), an active microtubule depolymerizer. **b** Left: average correlation between pairs of kymograph columns (Noco. $N = 38$ axons from four independent coverslips; DMSO $N = 22$ axons from eight coverslips) (Methods). Faster decay indicates a higher mobility of VAMP2:mCherry. Exponential functions with decay rate $\kappa$ were fitted to individual correlation curves (right). **c** Individual active transport (AT) events were segmented in kymographs and a motility index (Methods) was computed from them for each acquisition (same samples as **b**). **d** The velocity distribution of AT events (stairs) was well approximated by a bi-modal normal distribution in the control case (smooth blue line) (mean = 0.348 and 1.084 μm s⁻¹, DMSO $N = 1468$ AT events, Noco. $N = 856$ from axons in **b**) and by a single mode in the Noco. case (smooth red line) (mean = 0.438 μm s⁻¹). Dashed lines indicate the mean values of modes. **e**–**h** Similar imaging experiments and analysis when SVs were labeled by hyperkalemic stimulation with quantum dots (QD) targeted to synaptotagmin 1 (Syt1-IgG:QD). The median decay rates were 0.171 s⁻¹ (DMSO, $N = 12$ axons from seven coverslips) and 0.197 s⁻¹ (Noco., $N = 12$ axons from seven independent coverslips) (**f**), and median AT-based motility values were 0.0298 μm s⁻¹ (DMSO) and 0.0381 μm s⁻¹ (Noco.) (**g**). **h** Single Gaussian functions with mean = 537 μm s⁻¹ (DMSO, $N = 2871$ AT events) and 0.578 μm s⁻¹ (Noco., $N = 2862$ AT events) fit to AT velocity distributions (events from axons in **f**–**g**). Statistics: **b** right: DMSO median = 0.174 s⁻¹, Noco. median = 0.044 s⁻¹. Wilcoxon rank-sum (RS) test: $p < 10^{-4}$. **c**: DMSO median = $5.5 \times 10^{-3}$ s⁻¹, Noco. median = $0.9 \times 10^{-3}$ s⁻¹. RS test: $p < 10^{-6}$. **f**, RS test: $p > 0.08$. **g**, RS test: $p > 0.15$. Box plots: median, 25% and 75% percentiles (box) and extreme points (whiskers, excluding outliers).

kymographs of QD movements, prior nocodazole exposure spared traffic relative to control as measured by median autocorrelation of fluorescence along axons and pooled rate constants of decay (Fig. 1f). Likewise, in the analysis of individual AT events, nocodazole caused no drop in their rate of occurrence (Supplementary Fig. 5A), nor in aggregate motility (Fig. 1g) and spared the velocity distribution (Fig. 1h). The lack of effect of nocadazole on SV-QDs traffic was further confirmed by using FM 1–43 (Supplementary Figs. 5 and 7). The overall conclusion is

that QDs and FM 1–43 each highlight the same subset of the mixed population of vesicular structures: recycling SVs whose motility is spared by nocodazole-mediated microtubule disruption, and not the VAMP2:mCherry-bright vesicles that are presumed cytoplasmic transport packets whose motion is nocodazole-blocked.

**Actin polymerization is required for axonal SV traffic.** Finding no evidence for microtubule involvement in recycling SV

motility, we next considered the role of actin. Previous studies have implicated actin in extrasynaptic SV traffic[5,7,12], but its participation remains unclear because of seemingly discordant findings with different pharmacological interventions and monitoring methods. Using the Syt1-IgG:QD label, the same axonal segments were repeatedly imaged before and after pharmacological manipulations targeting actin. The agents differ markedly in a mode of action, yet each produced a striking decrease in SV motility (Fig. 2a, b). F-actin was likewise inhibited, either in the dynamics of polymerization (Fig. 2c, d) or in the filamentous structure itself (Fig. 2e, f).

The most telling intervention, brief exposure to latrunculin A at low concentration (0.1 μM, 3 min), causes buffering of free globular (G−) actin monomers and thus prevents G-actin recruitment to growing filaments[43]. This specifically interferes with elongation of dynamic actin filaments while sparing stable F-actin structures[24,25], as verified by phalloidin staining for F-actin (no difference in the area of phalloidin-positive structures between DMSO- and latrunculin A-treated neurons), whereas less mild treatment (30 min, 30 μM) produced cell-wide defects in the actin cytoskeleton (Fig. 2e, f). Yet, immediately after sub-micromolar latrunculin A treatment, aggregate AT motility was below control (median −83.1%), largely due a reduced AT rate (−71.2%) (Fig. 2b), reflected by only a few, short diagonal fluorescent trails amidst intense horizontal ones in kymographs (Fig. 2a). Individual AT events spared by latrunculin A showed a reduced median displacement length (−29.2%), mainly arising from reduced velocity (median −50.3%) insofar as the duration of AT was unchanged (+12.5%) (Supplementary Fig. 8).

We next examined transient actin structures using a fluorescent probe that binds to F-actin and quickly dissociates from G-actin: EGFP fused to the calponin homology domain of utrophin (Utr-CH:GFP)[25,44]. Transfected Utr-CH:GFP was expressed throughout whole neurons, including axons identified by thin, spineless morphology, ending in a stereotypical growth cone (Supplementary Fig. 9). Kymographs of axonal Utr-CH:GFP fluorescence displayed diagonal patterns of increase (Fig. 2c, left; confirmed by image enhancement, Supplementary Fig. 10D), reflecting aggregation of Utr-CH:GFP onto growing F-actin. Horizontal lines reflect stationary probe aggregates, arising from a combination of synaptic F-actin[45,46] and stable extrasynaptic actin patches[25]. Immediately after 3 min exposure to 0.1 μM latrunculin A, longitudinal actin polymerization in axons plummeted (Fig. 2c, right), confirming the previous work[25]. The rate of new filament outgrowth swiftly dropped (−67.4%), contributing to a sharp reduction (−85.2%) in the normalized cumulative length of actin polymerization (Methods). Compared to control, the spared polymerization after drug application appeared reduced in velocity (−20.3%), duration (−36.4%), and length (−56.8%) (Supplementary Fig. 8), in agreement with diminished availability of G-actin and impeded recruitment at F-actin's growing end. The parallel reduction of SV velocities suggests that the longitudinal elongation of actin might rate-limit SV motion.

Further tests were performed with jasplakinolide, which stabilizes actin filaments[47] and even promotes actin polymerization in vitro[47] and in synapses[46]. If SV traffic were supported by F-actin operating as a static rail for motor protein translocation, no reduction of mobility would be expected after F-actin stabilization. In contrast, the acute application of jasplakinolide (15 min, 10 μM, N = 5 coverslips) drastically reduced Syt1-IgG: QD AT rate (−94.6%) and abolished aggregate mobility (−96.9%) (Fig. 2b). In parallel, images of Utr:CH:GFP dynamics after jasplakinolide lacked detectable polymerization events (Fig. 2d), possibly resulting from extensive F-actin-stabilization[47] or depletion of G-actin at extrasynaptic segments

after synaptic enrichment in F-actin[46]. These findings affirmed the work of Darcy et al.[5], who found that jasplakinolide retarded SV-bound fluorescence recovery after photobleaching (FRAP) at individual synaptic boutons. For completeness, we also eliminated F-actin by prolonged application of a destabilizer, cytochalasin D[48] (30 min, 25 μg mL$^{-1}$), resulting in the disappearance of phalloidin staining (Fig. 2f). After treatment, AT of recycling SVs was hardly detectable: the rate of incidence fell sharply (−90.8%), as did SV mobility (−94.7%) (Fig. 2b), further corroborating the importance of F-actin for recycling SV motility. Overall, responses of SV traffic were well-aligned with those of actin polymerization for all interventions, suggesting a mechanistic connection.

**SV active transport is linked to local actin polymerization**. We next performed two-color imaging of fluorescently labeled vesicles and longitudinal filaments (Methods), comparing tracks of actin polymerization (green arrows and dashed lines) and SV motion (purple trajectories, Fig. 3a, b). Some SVs remained immobilized in F-actin dense regions, putative presynaptic structures (Fig. 3b, yellow arrowheads). Strikingly, initiation of AT events often aligned with the leading edge of actin polymerization and proceeded at a similar velocity (Fig. 3a, right). Multiple elongating filaments could cross the locus of a QD-labeled SV before it started to move (Fig. 3b right box, expanded in Fig. 3c top). Frequently, AT of SVs stopped while actin polymerization proceeded onward (Fig. 3a, b, bottom box; Fig. 3c, bottom). We also observed multiple secondary AT events, trailing behind a wavefront of F-actin polymerization that aligned with the initial AT initiation (Fig. 3c, bottom; 44% of AT-enabled SVs). For quantification, we compared Utr-CH:GFP fluorescence before and during AT near the SV locus (Fig. 3d). Tests against randomized pixel distributions revealed a significant increase of Utr-CH:GFP fluorescence in many (14/ 23) cases of AT (middle row). AT events arising first in AT trajectories coincided more frequently with local actin polymerization (8/11) than secondary events (6/12) (not significant). We considered whether first AT events rely on fresh actin polymerization, while secondary events do not because disengaged SVs can re-engage with preexisting filaments and then rely on motor-based transport[12]. This hypothesis pictures SVs hopping on, off, and on the same rails, implying that interruption of AT can occur without cessation of actin filament elongation. Indeed, Utr-CH:GFP fluorescence continued to increase ahead of SVs undergoing AT termination in half the cases where AT coincided with F-actin formation (Fig. 3d, bottom). Thus, SVs might stop moving either by dissociation from actin rails or because of F-actin elongation pauses.

We further analyzed AT events (N = 16) to quantify how closely actin polymerization and AT were spatiotemporally linked (Fig. 3e, f). We pinpointed the initiation of vesicle AT and measured its spatiotemporal separation from the actin polymerization wavefront. The start of SV AT lagged slightly behind the arrival of the F-actin tip only, by small margins in time (median = +1.1 s) and space (median = +0.27 μm) (Fig. 3e). AT velocity was generally slower than polymerization velocity in pairwise comparisons (Fig. 3f). Thus, slow F-actin elongation caps AT velocity when G-actin levels are latrunculin-depleted (Supplementary Fig. 8), but actin polymerization speed hardly limits SV motion under more physiological conditions. Our observations suggest instead that actin polymerization must bring tracks close to SVs for AT initiation and thus rate-limits its occurrence.

**Single SVs can undergo multiple rounds of active transport interspersed with diffusion**. Thus far we report motion of bright

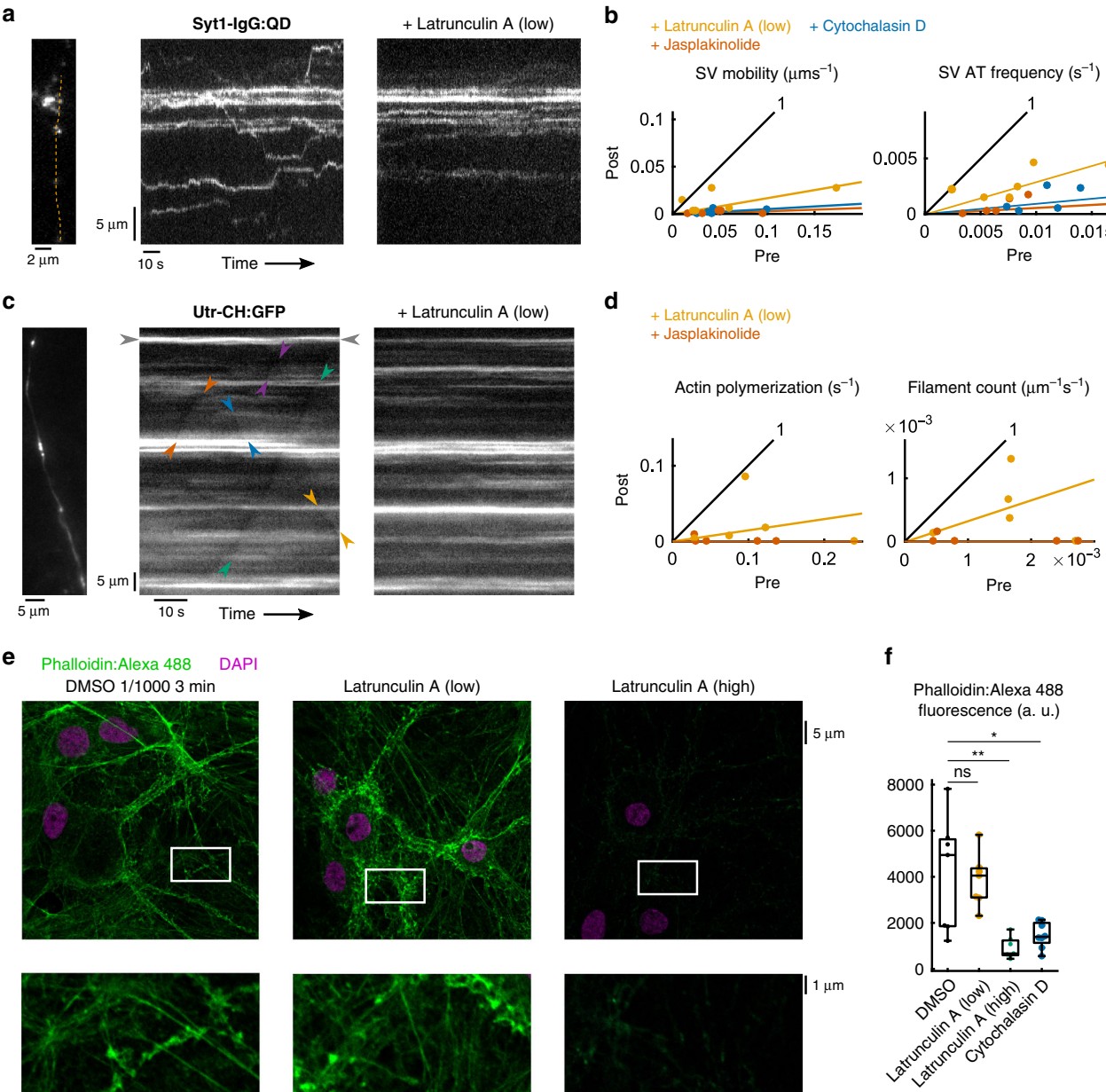

**Fig. 2 Inhibiting longitudinal polymerization of actin filament in axons impaired intersynaptic SV traffic.** We tested the effect of a low dose of latrunculin A (0.1 μM, 3 min) (low), a G-actin buffer, on the traffic of SVs when labeled with Syt1-IgG:QDs (**a**, **b**) and on neuronal actin (**c–f**). **a, b** Using paired experiments to minimize inter-coverslip variability, we compared the overall AT-based mobility and the rate of AT occurrence (SV AT frequency) (Methods) before (x axis) and after (y axis) treatment in the same independent coverslips (N = 6). Median changes: mobility −83.1%, p < 0.11; frequency −71.2%, p < 0.02 right-tail sign tests (S test). Similar experiments were performed with jasplakinolide (median change: mobility −96.9%, frequency −94.6%, p < 0.05, S tests for both) (N = 5 coverslips) and cytochalasin D (median change: mobility −94.7%, frequency −90.8%, p < 0.05 for both, S tests) (N = 5 coverslips). **c** In order to visualize actin polymerization along axons, cells were transfected so as to express the Utr-CH:GFP[44] (Supplementary Fig. 9). In control conditions (left) kymographs of Utr-CH:GFP showed horizontal (gray arrows) and diagonal lines (colored arrows) corresponding to fast longitudinal polymerization of actin. Arrows delineate the detectable initiation and termination points of polymerization for a few exemplar events. Right: same axon after treatment with latrunculin A (low). **d** Overall actin polymerization and the number of detected actin filaments (Methods) were quantified for both latrunculin A (low) (median changes: −85.2% mobility, −67.4% filament counts) (N = 5 coverslips) and jasplakinolide (median changes: −100% mobility, −100% filament counts) (N = 5 coverslips) in similar paired conditions as for Syt1-IgG:QD experiments (**a**, **b**). **e** Confocal microscopy images of cells that were subjected to latrunculin A (low) treatment immediately before fixation, permeabilization, and staining with DAPI and Phalloidin:Alexa 488 to visualize F-actin. Latrunculin A (high), corresponding to 30 min application at 30 μM as used in ref. [12], is also included. **f** Measurements of Phalloidin:Alexa 488 fluorescence as a proxy for total F-actin in cells immediately fixed after treatment with different actin-perturbing reagents (N = 7, 7, 7, and 8 independent coverslips, from left to right). Statistics in Supplementary Fig. 11. Box plots: median, 25% and 75% percentiles (box) and extreme points (whiskers, excluding outliers).

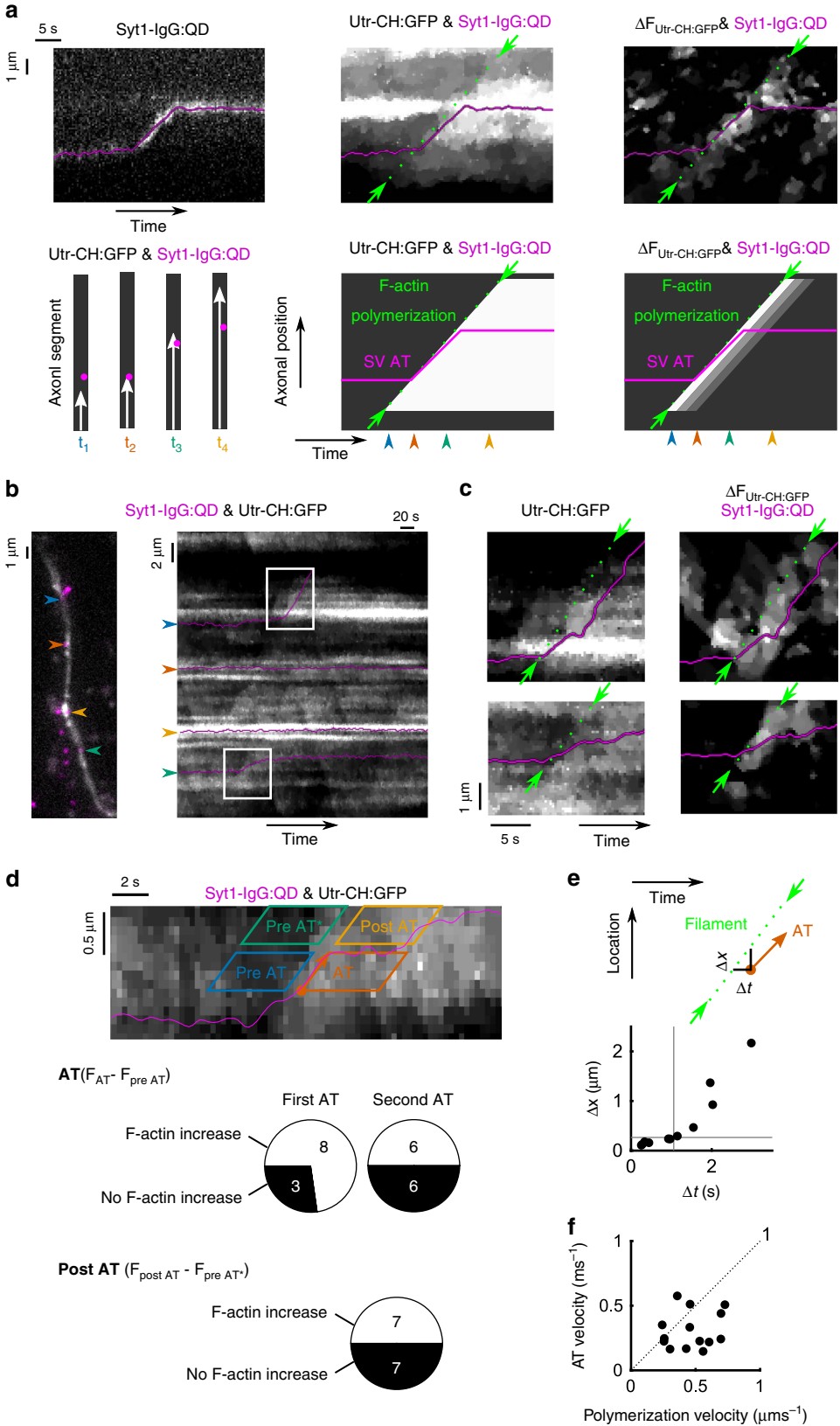

SV ensembles, moving as a group. To clarify individual SV dynamics, we used a sparse labeling approach to focus on single SVs (Fig. 4). Sub-pixel accuracy, 2D + T trajectories were automatically extracted (~30 nm mean localization error; Methods). AT epochs as brief as 0.25 s and as short as 125 nm were isolated by automatic segmentation via statistical testing against a

diffusion model (Fig. 4a, c). The median velocity was 0.96 μm s$^{-1}$ ($N = 91$ AT events, Fig. 4b), with a strong negative correlation between velocity and AT duration (Pearson correlation $= -0.58$, $p < 10^{-6}$, $t$-test). Velocity was rather homogeneously distributed (0–1.5 μm s$^{-1}$), while AT length and duration were right-skewed, with medians of 0.48 μm and 0.50 s, but reaching maximum

**Fig. 3 SV active transport coincided with local actin filament polymerization.** Dual-color imaging was used to simultaneously monitor polymerization of Utr-CH:GFP-labeled actin and QD-loaded SVs. Analysis of $\Delta F_{Utr-CH:GFP}$ (fluorescence at current time point minus average from $-6$ s to $-1$ s prior, positive values) enhanced polymerization signals (Supplementary Fig. 10). **a, b** In multiple instances, filament polymerization (green arrows and dotted lines) coordinated with AT in automatically extracted SV trajectories (magenta lines). **a** Bottom: idealized signals signifying coordinated AT and F-actin polymerization. **c** Zoomed-in view of boxes in **b**. **d** AT was automatically detected in QD trajectories and average Utr-CH:GFP fluorescence was computed at the locus of AT before ($-6$ s to $-1$ s before AT onset, blue parallelogram, "Pre AT") and during AT (from AT onset to 5 s after offset, red parallelogram, "AT"). The significance of fluorescence changes was computed by randomly shuffling pixels at the location of AT 1000 times and comparing actual values to the randomized distribution. A significance level of $\alpha = 0.05$, with Bonferroni correction for multiple tests, was used. Primary AT events ($N = 11$ events) that appeared first in SV trajectories were analyzed separately from later, secondary AT events (top charts) ($N = 11$ events) ($p < 0.27$, Fisher exact test when comparing the two groups). The same analysis was performed for fluorescence windows ahead ($+100$ to $+600$ nm) of the AT endpoint (green and yellow parallelograms) to determine whether termination of F-actin-associated AT was accompanied by cessation of polymerization ("Post AT" chart) ($N = 14$ events). **e** For AT events accompanied by actin polymerization ($N = 16$ events), we automatically fitted an edge-like linear profile in Utr-CH:GFP kymographs around the time of AT ($\pm 5$ s) in order to estimate the location of the filament tip. Top schematic: diagram of time delay ($\Delta t$) and spatial gap ($\Delta x$) between the AT initiation point (red dot) and the filament tip trajectory (green, dotted-line). Bottom: co-distribution of measurements, with gray lines representing median values. **f** Paired velocities of AT and actin polymerization, as the slopes of green and red lines in **e**. S test: $p = 0.038$ ($N = 16$ events). **d–f** Data were from the same eight independent coverslips.

values of 2.65 μm and 4.57 s (Fig. 4b). To determine whether AT along axons is influenced by proximity to presynaptic boutons, we combined sparse loading with QDs and labeling and exocytosis-driven unloading of FM 5–95 dye to mark presynaptic loci (Fig. 4c). Some QD-labeled SVs (>15%) remained nearly stationary within presynaptic loci over 2-min videos (Fig. 4c, top-left), as expected[5,12]. Some other SVs traveled along axons, often starting or stopping near a presynaptic vesicle cluster but not in its core (Fig. 4c, top-right); 30.1% of AT starts began <0.5 μm from the closest synapse, but none occurred within a synaptic cluster (Fig. 4d, bottom), as if escape from the SV coterie were a precondition. Some QD-labeled SVs traveled long distances (>5 μm) along axons[10] (Fig. 4a). In long trajectories, SVs moved past presynaptic clusters rather than crossing through them (Fig. 4c, bottom). There was no detectable correlation between kinetic properties of AT and proximity to the closest synaptic locus (velocity $\tau = 0.11$, $p = 0.35$; length $\tau = 0.21$, $p = 0.06$; duration $\tau = 0.10$, $p = 0.40$; Kendall's non-parametric correlation, $N = 39$). Thus, long-range transfer of SVs skirted around presynaptic clusters, consistent with EM images of synaptic boutons bulging away from the axonal shaft, leaving room for longitudinal translocation[5,49].

Constructing a >5 μm journey out of ~1 μm steps requires concatenation of multiple bouts of AT. The extrasynaptic SVs that engaged in long-range traffic displayed an excess of multiple AT events compared to a Poisson model wherein all SVs are equally likely to engage in AT (Fig. 4e). We observed 7 SVs with $\geq 4$ AT events ($p < 10^{-6}$) and 11 SVs undergoing 3 AT events ($p < 10^{-6}$), not the predicted zero or one. Thus, SVs that once underwent AT are more likely than overall to repeat. The proximity of F-actin could contribute to recurrent AT, seen when SVs seemingly hop off and back on the same actin track (Fig. 3c, bottom; Fig. 3d, top). Freedom from tethering might also contribute to heterogeneous dynamics, a possibility explored by examining the diffusional dynamics of SVs between AT steps. Analysis of mean squared displacement (MSD) vs. early time delay ($\Delta t$) revealed three major categories of SVs (Fig. 4f). SVs residing in synapses had rapidly saturating MSD curves, indicating sub-diffusive motion[50], as previously shown[11,51]. MSD curves of AT-disabled, extrasynaptic SVs also saturated, but at a higher level[11], indicating looser tethers. In contrast, AT-enabled SVs showed an almost linear relationship between MSDs and short $\Delta t$ between AT bouts, conforming to free diffusion. The differences in SV dynamics were confirmed by deriving a "diffusivity index" $\alpha$ (Fig. 4f, right). Thus, SVs undergoing long-range transport were relatively free of tethering between AT events, facilitating serial engagements with nearby actin filaments.

Repetitive AT of SVs could support directed re-distribution from a distinct locus to another, as for cytoplasmic packets from soma to distal axonal sites[16,28,38], endosomes[52], and autophagosomes[53,54] from neurite tips to soma. Instead, AT spread the superpool over the axon with little directional bias in the analysis of repetitive AT of SV clusters ($N = 327$). Patterns of AT conformed to a random-walk model wherein both directions of motion are nearly equivalent in probability and length (Supplementary Fig. 12). In contrast, microtubule-dependent AT of VAMP2:mCherry clusters was highly biased (~90% of movements occurring in the dominant direction) and well-fitted by directed re-distribution.

**PKA mobilizes SVs from synaptic clusters to increase traffic.** Disengagement of SVs from synaptic clusters[8,14,55–58] might provide a regulatory mechanism to control SV relocation by feeding extrasynaptic AT, thereby linking SV traffic to synaptic plasticity. We tested this hypothesis by activation of PKA, a well-studied intermediary of multiple neuromodulators, dopamine[59], serotonin[60], and brain-derived neurotrophic factor[61], which influence synaptic plasticity. Among many downstream actions, PKA can phosphorylate the vesicular adhesion molecule synapsin 1[62–64], leading to dissociation of SVs with synapsin 1 from synaptic clusters and subsequent dispersion[58,65–67], possibly by altering synapsin 1 liquid phases[57]. The deletion of synapsins greatly increases SV interbouton mobility seen as accelerated FRAP of synaptophysin-I:EGFP[68]. We asked whether PKA regulates SV availability for AT. PKA was activated by stimulating cyclic adenosine monophosphate (cAMP) production (10 μM forskolin, 10 min) after recycling SVs had been spontaneously labeled with Syt1-IgG:C488. Presynaptic clusters elongated by $+44\%$ (Fig. 5a). This matched the effect of synapsin 1 deletion[68] as if PKA-mediated phosphorylation and synapsin 1 removal resulted in a similar weakening of SV tethering. We next examined how this manipulation affected SV motion and F-actin dynamics. Forskolin treatment increased the overall motility of Syt1-IgG:QD-labeled SVs by $+172.6\%$ (median changes, Fig. 5b), but individual AT events were not faster (velocity, $-13.9\%$), nor longer-lasting (duration, $-8.21\%$), nor longer in distance (length, $-18.1\%$) (Fig. 5c and Supplementary Fig. 13). Similar effects were observed with okadaic acid (5 μM, 30 min), a phosphatase inhibitor that also disrupts SV clusters via enhanced synapsin 1 phosphorylation[64,66,69]: it increased median SV-QD mobility by $+370\%$ but also failed to boost unitary AT events (Fig. 5c and Supplementary Fig. 13). Because AT events were consistently elevated in frequency by forskolin (median $+49.3\%$) and okadaic acid (median $+126\%$) (Fig. 5b, right), we conclude that SV traffic

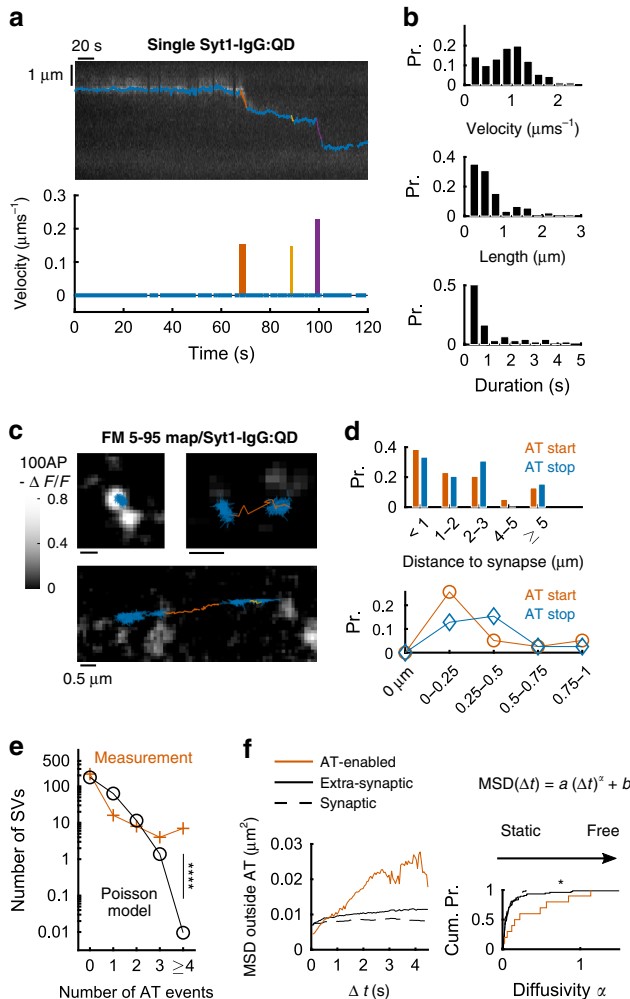

**Fig. 4 A distinct pool of diffusive SVs underwent multiple rounds of short active transport along axons.** Knowing that SVs take up at most one QD[30], we labeled single, isolated (10–20 per 50 × 50 μm field) SVs using a low (<0.5 nM) concentration of QDs during potent stimulation (45 mM K+, 90 s). In 2 min movies acquired at 20 Hz, individual trajectories were extracted automatically with sub-pixel accuracy and presumed AT events were detected by statistical testing (Methods). **a** Exemplar trajectory incorporating three AT events (red, yellow, and purple), interspersed with longer periods of relative immobility (blue). **b**–**f** Data were from the same eight independent coverslips. **b** Unitary properties of SV AT ($N = 91$ events). Velocity was 31% larger than for QD ensembles imaged at a lower frequency (Fig. 1), ($p < 10^{-6}$, RS test). **c** FM 5–95, a bulk marker of recycling SVs, loaded simultaneously with QDs and destained (grayscale) by electrical field stimulation (10 Hz, 10 s) to localize synaptic exocytosis. **d** Distribution of AT starting and ending points with respect to the closest synapse ($N = 113$ events). The distance of 0 μm corresponds to start/stop within an area of FM 5–95 release. **e** Number of AT events contained within 2 min trajectories of single SVs ($N = 77$ SVs). Measurements (red crosses) differ from predictions of a Poisson model (black circles) assuming an equal probability of AT occurrence between SVs. The Poisson model showed the low likelihood of observing 7 trajectories with ≥4 AT events and 11 trajectories with ≥3 AT events ($p < 10^{-6}$ in both cases). **f** Characterization of the diffusion of SVs exclusive of AT (blue periods in **a** and **c**). Mean squared displacements (MSD) for $N = 10$ AT-enabled SVs, $N = 76$ other extrasynaptic ("AT-disabled") SVs, and $N = 48$ synaptic SVs (left: median series). Right: cumulative distributions of the diffusivity index $\alpha$[50], for the same groups. Median values: AT-enabled SVs, $\alpha > 0.19$; extrasynaptic, AT-disabled, $\alpha = 0.05$; synaptic, $\alpha = 0.03$. Multiple comparison test: $p < 0.021$ (Kruskal–Wallis test by ranks). Synaptic vs extrasynaptic, AT-disabled SVs: $p > 0.79$, RS test; AT-enabled vs extrasynaptic AT-disabled: $p = 0.01$ (RS test); AT-enabled vs synaptic: $p = 0.005$ (RS test).

was modulated by elevation of the incidence of AT events, not of unitary properties. We next asked whether this arose from increased numbers of actin filaments supporting AT. However, no median change was detected with forskolin in total actin polymerization (−3.35%), the number of newly polymerizing filaments (−1.11%) (Fig. 5d), or unitary event properties (Fig. 5e and Supplementary Fig. 13). Seeing actin dynamics unchanged, we conclude that PKA regulates axonal traffic by acting as a gatekeeper for SV AT.

**NO-cGMP decrease SV AT and actin polymerization incidence.** We have addressed how actin polymerization is important for AT but not how actin modulation might control SV traffic. Previously, the arrest of mobile vesicles has been associated with local enhancement of vesicular release[36,70,71]. Antonova et al. found that presynaptic vesicle protein clusters increase following long-term potentiation (LTP)-inducing protocols in hippocampal cultures[36,70,71]; Wang et al.[36,71] suggested presynaptic roles for retrogradely diffusing NO, cyclic GMP-dependent protein kinase (cGK), RhoA and actin polymerization as:

↑NO→↑presyn cGMP→↑cGK activation→↓RhoA activity→↓actin polymerization.

Our finding that vesicle motion depends on F-actin elongation provides a mechanism by which retrograde regulation of actin polymerization could impede SV traffic and thus engender presynaptic clustering of vesicles as previously observed[36]. Accordingly, we asked whether elevation of NO or activation of cGK reduced actin polymerization and

consequently SV traffic (Fig. 6a). Application of a transient NO donor, DEA-NONOate (3 μM), quickly reduced total polymerization of axonal actin (−44.6% average change) and new filament abundance (Fig. 6b). Upon removal of DEA-NONOate polymerization recovered promptly and fully (+4.05% average change). NO stimulates guanylate cyclase to elevate cGMP. Application of 8-br-cGMP, a non-hydrolyzable cGMP compound, mimicked DEA-NONOate: −34.2% average decrease in total polymerization length and −36.9% average decrease in the number of new filaments. The effect of 8-Br-cGMP application was not reversible, consistent with its durable activation of protein kinase G (PKG)[72]. We found no significant change in velocity, length, and duration of AT segments with either DEA-NONOate or 8-Br-cGMP (Fig. 6b and Supplementary Fig. 14). We then tested whether NO or cGMP would affect SV traffic. Brief application of DEA-NONOate (3 μM, 4 min) reduced SV mobility and AT count by 40–50% with partial reversal after 5 min wash in drug-free solution. Likewise, exposure to 8-br-cGMP (50 μM, 4 min) produced a similar reduction in SV-QD mobility (average mobility, −40.7%; average AT frequency, −34.0%). Once again, this action was not reversed even after 10 min wash (average mobility −59.4%, average AT frequency −51.4%). Moreover, unitary properties of AT remained unchanged and did not contribute to the lowered mobility (Fig. 6c and Supplementary Fig. 14). In summary, actin polymerization and SV motility showed highly correlated responses to manipulations of NO-cGMP signaling.

A likely downstream mediator of NO-cGMP is RhoA, a rho family GTPase that controls actin polymerization in various systems[73]. RhoA triggers two main effector cascades (Fig. 7a), one mediated by the diaphanous-related formin mDia1. Favoring this

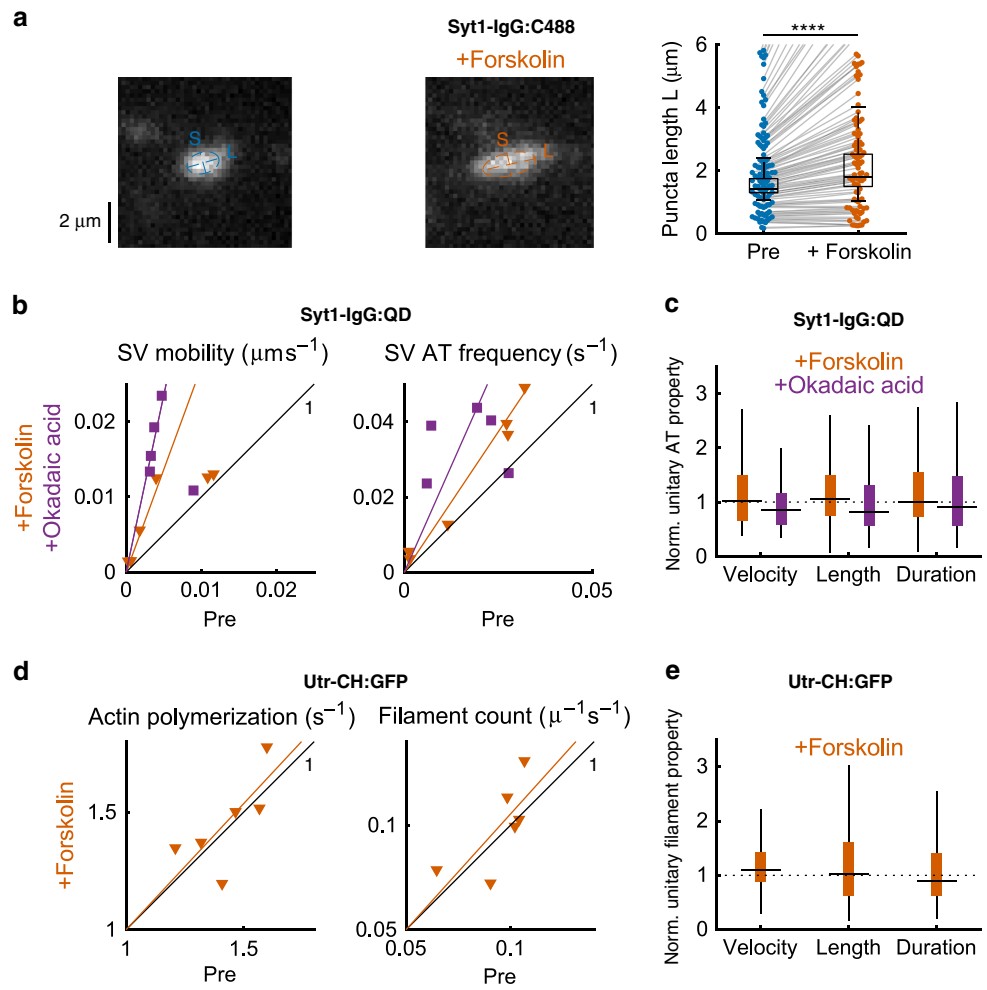

**Fig. 5 PKA regulated the rate of AT occurrence by tuning SV availability. a** SVs were extensively stained with a fluorescent antibody against a luminal epitope of synaptotagmin 1 (Syt1-IgG:C488) by spontaneous SV fusion and recycling during a 2–3 h incubation period. Stable large clusters of fluorescence presumably corresponded to synaptic clusters of vesicles (left). Middle: the same synapses were examined after a 10 min treatment with 10 μM forskolin to increase cAMP production and activate protein kinase A (PKA). Punctum shape was characterized by fitting rotated 2D Gaussian functions to images and calculating the long (L) and short (S) axes of the shape before and after exposure to forskolin (right). L average +44%; median +19%; S test: $p < 10^{-4}$ ($N = 91$ puncta from two coverslips). The same pharmacological manipulation was applied to SVs loaded with Syt1-IgG:QDs and overall SV mobility, AT frequency (**b**), and unitary properties of AT (**c**) were characterized and compared in a pairwise fashion. The frequency of the AT events was consistently elevated by forskolin (+49.3% median increase, $p < 0.02$, S test, $N = 6$ coverslips). Multivariate analysis of variance (mANOVA) including velocity, length, and duration of AT events: $p < 10^{-6}$ (pre, $N = 681$ events; post, $N = 1025$ events). Velocity median change −13.9%, $p < 10^{-6}$, RS test; pre, $N = 681$ events, post, $N = 1025$ events from the same 6 coverslips. Duration median change −8.21%, $p > 0.07$, RS test. Length median change −18.1%, $p < 10^{-6}$, RS test. Okadaic acid (purple) was also used to inhibit a range of phosphatases and resulted in a similar increase of mobility ($p = 0.031$, S test, $N = 5$ coverslips) and of AT frequency (median +126%, $p = 0.19$, S test, 4/5 increase), without changes of unitary properties mANOVA $p > 0.48$ (pre, $N = 421$ events; post $N = 1074$ events). **d**, **e** Actin polymerization was imaged and characterized before and after forskolin treatment in a paired experimental paradigm. No change was detected in **d** total actin polymerization (−3.35% median change, $p > 0.34$, S test; $N = 6$ coverslips), the number of newly polymerizing filaments (−1.11% median change, $p > 0.65$, S test), or **e** unitary event properties (mANOVA $p > 0.05$; pre, $N = 180$ events; post $N = 327$ events). Box plots: median, 25% and 75% percentiles (box) and extreme points (whiskers, excluding outliers).

pathway of RhoA involvement, the pharmacological block of formins inhibits longitudinal actin polymerization along axons of hippocampal neurons[25]. To test specifically for participation of mDia1, we co-expressed a dominant-negative variant of mDia1 (mDia1-DN:YFP) with a red sensor for polymerized actin (Lifeact:tdTomato) instead of UTR-CH:GFP (Supplementary Fig. 15). mDia1-DN expression reduced total longitudinal polymerization (median −58.3%) and the number of filaments (median −52.1%) (Fig. 7f), while spared filaments were moderately affected (Supplementary Fig. 15). The other possible cascade initiated by RhoA proceeds through ROCK, whose

activity promotes filament elongation (lower branch, Fig. 7a)[36]. Inhibition of ROCK with Y27632 (2 h, 10 μM) reduced total actin polymerization (median −39.2%) and the count of new actin filaments (median −38.0%) (Fig. 7g), while remaining filaments were mostly unaltered (Supplementary Fig. 16). These observations indicated that both mDia1- and ROCK-dependent branches promote actin polymerization in hippocampal axons. By controlling both pathways, RhoA is poised to regulate longitudinal F-actin assembly. Accordingly, we next tested whether RhoA inhibition with the C3 transferase exoenzyme (C3)[74] would concomitantly affect actin dynamics and vesicle trafficking. C3

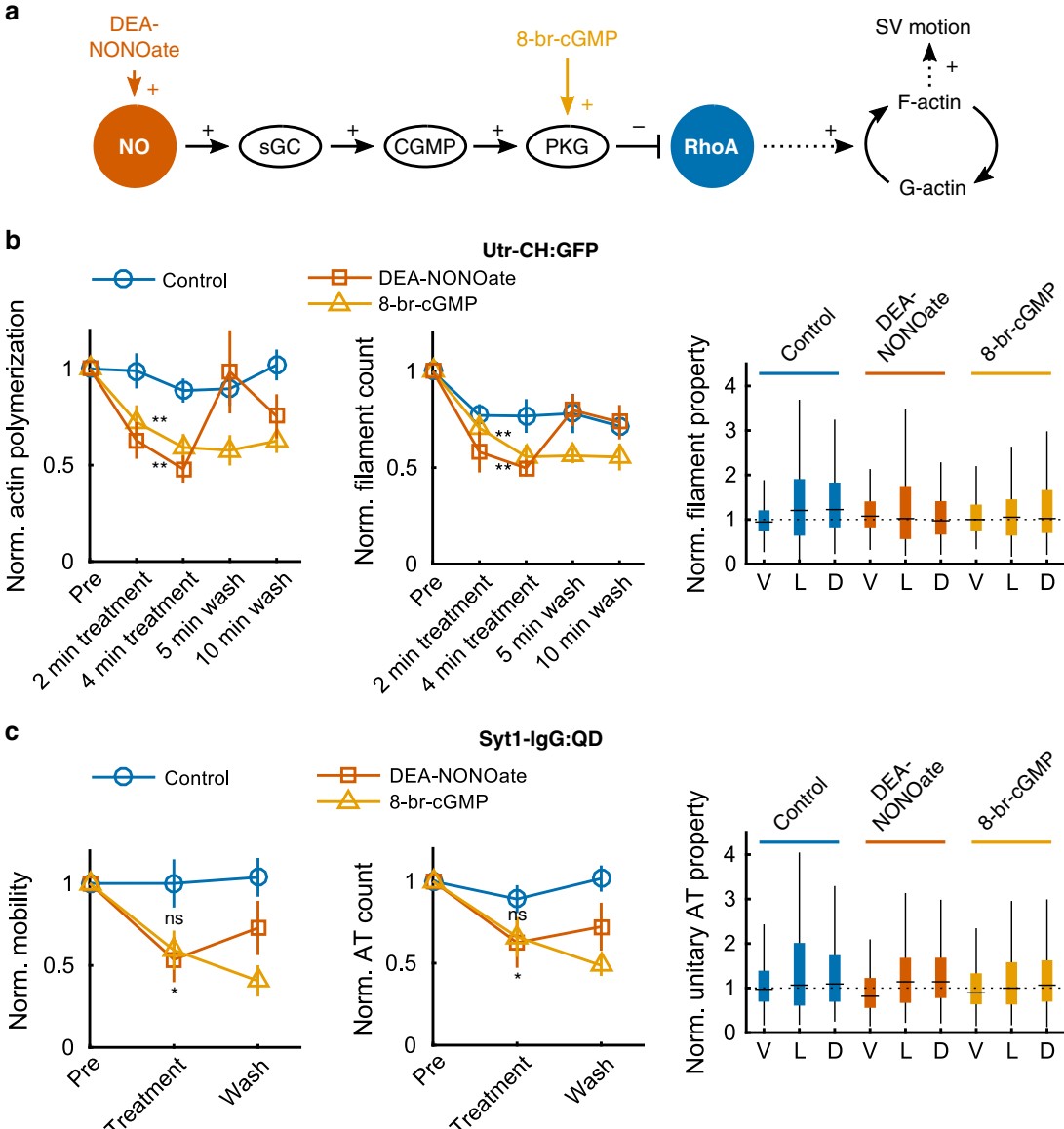

**Fig. 6 Inhibiting NO activity impaired both longitudinal actin polymerization and SV traffic along axons.** The molecular pathway described in **a** was tested to determine whether it controls dynamic actin filaments and SV traffic along axons. **b** Utr-CH:GFP-expressing cells were treated with the transient NO donor DEA-NONOate (N = 8 coverslips), a sham solution (N = 10 coverslips), or a non-hydrolyzable form of cGMP (N = 7 coverslips), and longitudinal actin polymerization was quantified by time-lapse microscopy. Results from two movies pre-treatment were averaged to obtain a robust baseline (Pre) for individually normalizing results from each coverslip. **c** Similar manipulations when imaging the traffic of SVs labeled with Syt1-IgG:QDs (control, N = 5 coverslips; DEA-NONOate, N = 5 coverslips; 8-br-cGMP N = 4 coverslips). Only one acquisition was performed for pre-treatment, (4 min) treatment, and (10 min) wash in order to reduce the phototoxicity due to the ultra violet illumination light. Analysis of unitary properties for all conditions (**c** right and Supplementary Fig. 14) shows that the reduction of polymerization for DEA-NONOate and 8-br-cGMP treatments cannot be explained by shorter actin filaments. Statistics: **b** DEA-NONOate: total polymerization −44.6% average change, decrease for 8/8 coverslips, $p < 10^{-2}$, S test; count of new filaments: −46.1% mean change, decrease for 8/8 coverslips, $p < 10^{-2}$, S test. 8-br-cGMP: −34.2% average total polymerization length ($p < 10^{-2}$, S test) and −36.9% average number of new filaments ($p < 10^{-2}$, S test) (N = 7 coverslips). Measurement at 2- and 4-min treatment were averaged. mANOVA for unitary filaments including velocity (V), length (L) and duration (d) for DEA-NONOate: $p > 0.84$ (pre, N = 226 events; post, N = 218 events) and 8-br-cGMP: $p > 0.79$ (pre, N = 267 events, post, N = 172 events). **c** DEA-NONOate SV AT mobility and frequency: decrease for 5/5 coverslips, $p = 0.031$, S test for each. 8-br-cGMP: average mobility −40.7% (decrease for 4/4 coverslips, $p = 0.062$, S test); average AT frequency −34.0% (decrease for 4/4 coverslips). mANOVA for unitary events for the DEA-NONOate: $p < 10^{-6}$ (pre, N = 543 events; post, N = 216 events). Velocity median −11.0% ($p < 0.02$, RS test), length median +24.0% ($p < 0.005$, RS test), duration median +32.0% ($p < 10^{-6}$, RS test). The same analysis for 8-br-cGMP: mANOVA $p = 0.084$ (pre N = 853; post, N = 329). Box plots: median, 25% and 75% percentiles (box), and extreme points (whiskers, excluding outliers).

application (2 h, 15 µg mL$^{-1}$) disrupted F-actin (Supplementary Fig. 11G) and reduced dynamic actin polymerization (median −31.7%) and the abundance of new actin filaments (median −32.9%) in time-lapse imaging of Utr-CH:GFP (Fig. 7b, c). C3 treatment spared unitary properties of polymerizing filaments

(Fig. 7c, right), like DEA-NONOate and 8-br-cGMP. Importantly, C3 application also reduced SV motility along axons (median −61.0%) and the frequency of occurrence of AT (median −65.1%) (Fig. 7d, e), while sparing unitary AT events (Supplementary Fig. 16B), paralleling C3 effects on F-actin in all

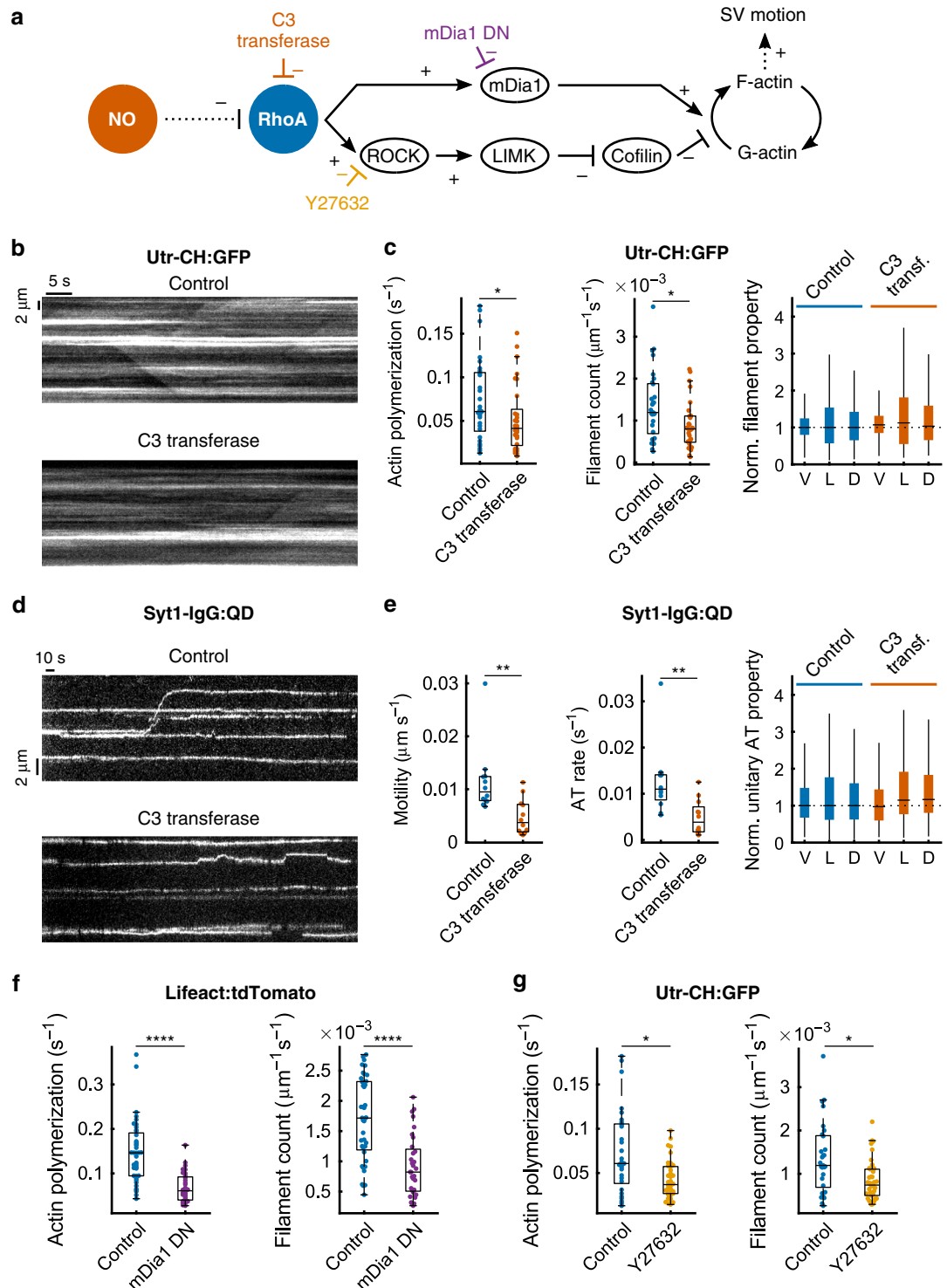

respects. Thus, the block of RhoA fully mimicked the effects of suppressing NO or cGMP signaling, supporting its proposed relay action is impeding both F-actin polymerization and SV motion.

## Discussion

Focusing on presynaptic vesicles that undergo activity-dependent fusion/retrieval, we reveal that their axonal movements are longitudinally bidirectional, depend on actin, and do not require nocodazole-sensitive microtubules. SV AT was not detectably affected by nocodazole or colchicine, a microtubule-disrupting agent with a distinct mechanism (Supplementary Fig. 17). In contrast, the motion of vesicular transport packets[25,28,29], labeled via VAMP2/synaptobrevin but not endocytosed markers, was confirmed to be strongly directionally-biased and microtubule-dependent (Supplementary Fig. 12). These distinctions extend current views of axonal membrane traffic[16–18] that emphasize microtubules and recall photobleaching experiments that first implicated actin in interbouton SV motion[5].

To clarify the structural determinants of SV dynamics, we used two-color imaging to monitor the motion of recycling vesicles

**Fig. 7 Inhibiting RhoA activity impaired both longitudinal actin polymerization and SV traffic along axons.** The molecular pathway in **a** was explored to determine the role of RhoA and its effectors in controlling longitudinal actin polymerization and SV traffic. **b** Exemplar kymographs of actin polymerization, assessed by Utr-CH:GFP fluorescence, after application of the C3 transferase exoenzyme or in a control, separate coverslip. **c** Quantification of C3 transferase treatment ($N = 29$ movies from three coverslips) and in control sister coverslips ($N = 29$ movies from three coverslips). **d, e** Median exemplar kymographs of Syt1-IgG:QD images and quantitative analysis of SV traffic for the same manipulations. Analysis of unitary filaments and AT events (V: velocity, L: length, d: duration) are detailed in Supplementary Fig. 16. **f** Cells were co-transfected with a dominant-negative construct for mDia1 (mDia1-DN) and a red-shifted probe for F-actin (Lifeact:tdTomato) (Supplementary Fig. 15) (wild type, $N = 40$ movies from 6 coverslips; mDia1-DN, $N = 32$ movies from 7 coverslips). **g** Actin polymerization was monitored when the Rho activated kinase (ROCK) was inhibited with Y27632 (10 μM, 2 h) (control $N = 29$ movies same as **c**; Y27632 $N = 38$ from 4 coverslips). Statistics: **c** median dynamic actin polymerization $-31.7\%$, $p = 0.04$, RS test. New actin filaments count median $-32.9\%$, $p = 0.015$, RS test. mANOVA including $V$, $L$, and $D$: $p < 10^{-4}$. Velocity median $+7.05\%$ ($p > 0.11$, RS test), length median $+12.4\%$ ($p > 0.13$, RS test), duration median $+3.30\%$ ($p > 0.23$, RS test) (C3 transferase $N = 209$ filaments, control $N = 530$ filaments). **e** Median AT mobility ($-61.0\%$, $p < 10^{-2}$, RS test; median AT occurrence rate $-65.1\%$, $p < 10^{-2}$, RS test) (DMSO, $N = 12$ movies from 7 coverslips; C3, $N = 12$ from 5 coverslips). mANOVA: $p < 10^{-6}$. Velocity median $-2.78\%$ ($p = 0.024$, RS test), length median $+14.9\%$ ($p < 10^{-6}$, RS test), duration median $+16.7\%$ ($p < 10^{-6}$, RS test) (C3 transferase $N = 931$ events, control $N = 1898$ events). **f** Median total longitudinal polymerization $-58.3\%$, $p < 10^{-6}$, RS test; median number of new filaments $-52.1\%$, $p < 10^{-5}$, RS test. **g** Median total actin polymerization $-31.7\%$, $p = 0.04$, RS test; median abundance of new actin filaments $-32.9\%$, $p = 0.015$, RS test. Box plots: median, 25% and 75% percentiles (box) and extreme points (whiskers, excluding outliers).

while simultaneously tracking actin elongation[25]. The initiation of vesicle movement corresponded spatiotemporally with the arrival of growing actin structures. Rather than actin-driven propulsion[75], our experiments support repeated binding and unbinding of SV to the newly laid actin track together with motor-mediated processive motion (Fig. 3), in line with myosin V inhibition experiments[12]. F-actin nucleation and elongation preceded SV transport, rather than being triggered by it as in Rab11a transport in oocytes[76]. The formation of fresh actin tracks dictated the rate of SV AT occurrence (Figs. 2, 6, and 7) and the speed of filament elongation put a ceiling on vesicle velocity. Actin filament polymerization and SV AT were slowed to a similar degree by exposure to latrunculin A (Supplementary Fig. 8), or by manipulating NO-cGMP-RhoA signaling. Unlike actin-based transport of axonal organelles in invertebrate axons, which persists after phalloidin-based actin stabilization[19,21–23], actin stabilization abolished SV traffic (Fig. 2)[5]. Beyond differences in how actin polymerization is controlled or exploited[20,22,23,25], a common feature is that vesicle motion depends on actomyosin interaction[12,22,23,77]. Continuously tracking QD-containing SVs, we affirmed that they can travel for long distances along axons[15], doing so by combining episodes of directed AT, interspersed with diffusive motion. AT frequently started and ended at the periphery of synaptic clusters, but never in their core, confirming that AT reflected extra- rather than intra-synaptic mobility. Others have implicated SV-actin interactions in sub-diffusive motion[7,51,78] (Fig. 4), oscillating states[11], and mobility within synaptic clusters, possibly via acto-myosin2 complexes[11,27,79,80]. In contrast, a major role for myosin II in extrasynaptic SV mobility was not apparent from a pharmacological intervention (Supplementary Fig. 18). Myosin V, a processive motor, seems better suited to carry out repetitive AT along axons[12]. Interspersed diffusive pauses began when vesicles ceased to associate with nearby actin structures; directed motion resumed both without and with the arrival of a new growing actin structure as if temporarily disengaged vesicles can either re-engage with the original track or with a fresh one.

SVs moved bi-directionally, without an obviously preferred direction under basal conditions (Supplementary Fig. 12), enabling axonal re-distribution of SVs among synaptic sites. Dynamic actin filaments reported by others[25] and by us are suited for this function in growing in both axonal directions without strong bias[25]. In contrast, axonal microtubules offer greater stability and singular polarization[81], befitting efficient and uni-directional fast transport between distant, pre-determined sites (soma to remote synaptic sites or vice-versa).

By revealing checkpoints for SV trafficking, our work suggested scenarios by which neuromodulation could regulate interbouton exchange in association with learning[82,83]. Intracellular cAMP can be elevated by dopamine-, serotonin- or activity-dependent activation of adenylate cyclase. cAMP elevation with forskolin enriched vesicles in the periphery of synaptic clusters, likely mediated by PKA phosphorylation of vesicular synapsin 1[58], liberating vesicles from synaptic confinement[65–67]. We also considered alternatives such as PKA-mediated modulation of myosin V or myosin light chain kinase[84] (but see Fig. 5c and Supplementary Fig. 18). Freeing vesicles from cytoskeletal constraints with PKA would make more vesicles eligible for interbouton trafficking, enlarging the vesicular "superpool"[8,15,58]. Indeed, we observed that PKA stimulation increased the abundance of AT events without changing actin elongation or AT unitary properties.

Postsynaptic NMDA receptor activation locally enhances post- and presynaptic function in certain forms of LTP. Presynaptic enhancement is reflected by elevation of mini frequency[70], of uptake of presynaptic vesicular tags[85], of the number of synaptophysin-immunopositive puncta[36] and of interbouton vesicle traffic[9]. Wang et al. showed that SV proteins accumulate at potentiated synapses by means of signaling via NO, cGMP, PKG, and RhoA. Our observations (Figs. 6 and 7) provide a way to link plasticity-inducing signals to the recruitment of presynaptic vesicular resources. We propose that retrograde signaling by NO drives a local cascade of NO → ↑cGMP → ↑PKG → ↑RhoA[86,87] to inhibit actin elongation and stall SV transport, enabling recruitment of stalled vesicles into synaptic pools at presynaptic sites destined for potentiation. We found that raising NO or cGMP, key players in retrograde signaling, dampened actin filament elongation, and vesicle transport, involving both mDia1- and ROCK-dependent branches of signaling between RhoA and actin dynamics.

Such local recruitment of SVs could operate in tandem with a generalized PKA-dependent mobilization of SVs[58] to concentrate SVs at potentiated postsynaptic sites, thus coordinating pre- and postsynaptic capabilities. By supporting short-haul SV traffic, actin-based SV movement facilitates local retrograde control, serving an altogether different function than microtubule-based, long-range, movements of transport vesicles.

## Methods

**Primary cultures of hippocampal neurons.** Hippocampal neurons were cultured from postnatal day 0 male and female Sprague-Dawley rat pups (1–6 h after birth). Both hippocampi were isolated in ice-cold modified Hank's balanced salt solution (HBSS; 21-021-CV, Gibco ThermoFisher Scientific, Waltham, MA; added 1 mM

HEPES, pH 7.4) containing 20% heat-inactivated fetal bovine serum (FBS; S1115OH, Atlanta Biologicals, GA). They were cleaned from meninges and dentate gyruses were dissected out. Hippocampi were divided in 3 to 4 pieces before being washed twice in ice-cold modified HBSS with 20% FBS. Pieces were then digested for 8 min in 1 mL papain solution (20 U Papain, LK003178, Worthington, Lakewood, NJ; 1 mL HBSS; 1050 U DNAse I, D5025, Sigma-Aldrich, St Louis, MO) at 37 °C. Digestion was stopped by adding 5 mL of ice-cold modified HBSS with 20% FBS and pieces were washed multiple times with this solution. Hippocampal pieces were then placed in DNASe I-containing HBSS (350 U) and the tissue was dissociated using three fire-polished Pasteur pipettes of decreasing diameter. The cell suspension was pelleted, resuspended in prewarmed cultured medium (NbActiv4, BrainBits, Springfield, IL) supplemented with 10% FBS and plated on 10 mm diameter coverslips coated with poly-D-lysine (P7405, Sigma-Aldrich). After 2 days, the culture medium was exchanged for FBS-free NbActiv4 medium with 0.25 μM Ara-C (C6645 Sigma-Aldrich) to halt glial cell proliferation. 40% of the culture medium was exchanged for fresh NbActiv4 at DIV 8–9 and experiments were performed at DIV 12–16. All procedures involving animals were approved by the Institutional Animal Care and Use Committee at the New York University Langone Medical Center (NYULMC), and in accordance with guidelines from the National Institutes of Health.

**Plasmids and transfection**. We used the following plasmids: VAMP2:mCherry was custom-made by Damon Poburko and Yulong Li, YFP-mDia1FH2ΔN (mDia1-DN) was a gift from Arthur Alberts (Addgene plasmid #25419)[88], Lifeact-7:tdTomato was a gift from Michael Davidson (Addgene plasmid #54528)[89], Utr-CH:GFP was a gift from William Bement (Addgene plasmid #26737)[44].

Plasmid transfections were performed at DIV 9 using the following protocol. Each coverslip was incubated with 0.3 μL Lipofectamine® 2000 (ThermoFisher Scientific) reagent and 1 μg plasmid DNA for 4 h. The transfection medium was then replaced by a conditioned culture medium free of plasmid DNA and Lipofectamine reagents. To decrease the toxicity of the expression of Utr-CH:GFP and Lifeact-7:tdTomato proteins expression the expression level was lowered by using 0.1 μg plasmid DNA per coverslip. For dual-color imaging (Fig. 3) transfection was performed at DIV 7 using a standard calcium phosphate-based method adapted to neuronal cultures[90].

**Pharmacology**. Drugs were purchased, prepared, and stored as described in Supplementary Table 1. Each DEA-NONOate aliquot was thawed and diluted to the appropriate concentration in the experimental solution at 37 °C immediately before application. Other reagents were thawed and kept at 4 °C for up to 24 h.

**Time-lapse imaging of vesicle traffic and actin polymerization**. Microscopy setup: A customized microscopy setup was used for time-lapse imaging of VAMP2:mCherry, FM dyes dynamics, quantum dots (QDs) traffic, and actin polymerization as seen with Utr-CH:GFP and Lifeact-7:tdTomato. An Eclipse Ti-S (Nikon, Japan) base microscope was equipped with an oil immersion apochromat objective (Nikon) (×100, NA 1.4), an electron-multiplying (EM) CCD camera (Ixon+, Andor, N. Ireland), and sets of band-pass excitation and emission filters adapted to the spectra of excitation lights and fluorescent dyes. For excitation, we switched between three laser lines (405, 488, and 532 nm, Crystalaser, Reno, NV), neutral filtered for optimal signal quality while limiting phototoxicity. A Uniblitz mechanical shutter (Vincent Associates, Rochester, NY) was controlled by the Andor acquisition software. 405 nm light was used to illuminate QDs alone. The 488 nm light was used to illuminate GFP derivatives and FM dyes. VAMP2:mCherry and Lifeact-7:tdTomato were excited with the 532 nm light. At the input, a custom beam expander was assembled with a set of lenses to illuminate the sample uniformly. At the output, another custom expansion system was assembled to reach the Rayleigh criterion for signal sampling. The final pixel size was 96 nm. The microscope was surrounded by a custom-made Plexiglas environmental chamber in which temperature was stabilized at 37 °C, except otherwise indicated, by using a closed-loop air heating system (World Precision Instruments, Sarasota, FL).

Dual-color imaging of Utr-CH:GFP and QDs: To simultaneously image Utr-CH:GFP and QDs dynamics we modified the expansion lens system at the output to include a dichroic mirror which split red and green fluorescence into two separated optical paths (565 nm cutoff, Beamsplitter T 565 LPXR, Chroma Technology Corp., Bellows Falls, VT). Each path was carefully adjusted with mirrors and lenses so that they reached separate halves of the camera sensor array, side-by-side, and with the same focus. The overlap of both green channels was then calibrated with sub-pixel accuracy by using fixed, standard samples.

Experimental solutions: Coverslips layered with cells were mounted in an imaging chamber with a glass bottom (Harvard apparatus, Holliston, MA) and continuously superfused with a 37 °C modified Tyrode's solution containing: 120 mM NaCl, 4 mM KCl, 2 mM CaCl₂, 2 mM MgCl₂, 10 mM glucose, 10 mM HEPES and adjusted to 280 mOsm and pH 7.4 (using NaOH and HCl). When imaging QDs with blue light, 5 mM DL Dithiothreitol (Sigma-Aldrich, D9779) was added to the solution to cope with phototoxicity. For QD imaging, 1 μM of the BHQ-3 (BHQ-3001-5, Biosearch Technologies. Petaluma, CA) or 5 μM trypan blue (Sigma-Aldrich) were also used to quench the fluorescence of extracellular QDs. The "high potassium solution" for dye

and QD loading/destaining consisted in 85 mM NaCl, 45 mM KCl, 2 mM CaCl₂, 2 mM MgCl₂, 5 mM glucose, 5 mM HEPES. All solutions contained 5 μM NBQX and 50 μM D-AP5 (Ascent Scientific Ltd).

Time-lapse imaging configuration: We adopted different imaging strategies depending on the experiment goal, the fluorophore characteristics, and the phototoxicity induced by imaging the same biological object multiple times. To image QD traffic at the level of synaptic vesicle (SV) clusters we acquired 2 min long movies with 3 Hz imaging rate (405 nm laser, 80 ms exposure). For single SV imaging, we increased the acquisition rate to 20 Hz with continuous light exposure for 2 min and we changed the field of view in between movies. For repeated Utr-CH:GFP imaging, we used a 488 nm laser with 5 Hz acquisition rate and continuous imaging for one minute. Lifeact-7:tdTomato was imaged with similar settings, but using a 532 nm excitation laser. VAMP2:mCherry was imaged for 2 min at 3 Hz with 100 ms exposure to a 532 nm light. Dual-color imaging was at 3 Hz, 80 ms exposure to a 488 nm light, for 2 min. Excitation light intensity was adjusted with neutral density filters to optimize signal quality and reduce phototoxicity, but the configuration was kept fixed in between coverslips from the same experiment.

**FM dye loading of SVs**. After 10 min habituation of cells to the experimental solution, a 25 μL drop of high potassium solution with 10 μM FM 1–43 or FM 5–95 (Biotium, Fremont, CA, Cat. No. 70020 and 70028) dissolved in DMSO (final < 1/1000) was placed on the coverslip for 90 s. Then, the solution was diluted 4× with normal Tyrode's solution containing 10 μM FM dye, and after an extra 90 s cells were extensively washed with dye-free Tyrode's solution. This high potassium-based protocol extensively stains the recycling pool of SVs[91]. FM 1–43 puncta traffic was imaged at 3 Hz for 2 min (50 ms exposure, 488 nm).

**QD labeling of SVs**. Two to four hours prior to experiments, biotinylated IgG binding to the lumenal domain of synaptotagmin 1 (105 311BT, Synaptic System, Göttingen, Germany) were added to the culture medium of cells (0.3 μg mL⁻¹). At the time of experiments, non-bound antibodies were extensively washed with Tyrode's solution and 2% bovine serum albumin (BSA)(Sigma-Aldrich) was then added for 5 min to prevent non-specific QD binding to cell membranes. Streptavidin-coated QDs with 655 nm peak emission (ThermoFisher Scientific, Cat. No. Q10123MP) were applied in a high potassium Tyrode's solution containing 2% BSA for 90 s. QD concentration was 2–4 nM, except for a single vesicle experiment for which concentration was lowered to 0.2–0.5 nM. Cells were washed at 37 °C before starting imaging experiments.

**Electrical stimulation**. Field stimulation was performed using custom-made, parallel platinum electrodes connected to a Grass Stimulator (SD9, Grass Technologies, West Warwick, RI) and inserted into the imaging chamber.

**Immunocytochemistry**. Fixed F-actin staining: After cells were incubated in Tyrode's solution and subjected to the proper pharmacological treatment, they were fixed for 10 min using an ice-cold phosphate buffer saline (PBS) solution containing 4% paraformaldehyde (PFA) and 40 mg mL⁻¹ sucrose. 0.1 % Triton X-100 in PSB was then used for 5 min at room temperature to permeabilize cells. After incubation for 30 min in a blocking PBS solution (1% bovine serum albumin, BSA), 5 μL of Alexa Fluor 488 Phalloidin (A12379, ThermoFisher Scientific) (6.6 μM in methanol) was diluted to 1 mL of PBS and applied to cells for 45 min. Coverslips were mounted on a glass slide in a DAPI containing medium (Prolong gold, P36930, ThermoFisher Scientific) after three washes with PBS.

Fixed microtubules immunolabeling: For fixed cells α-tubulin immunolabeling, we adapted a protocol published online by the Mitchinson lab (Harvard University, USA). Briefly, non-polymerized tubulin was extracted by immersing cells for 30 s in a microtubule-stabilizing buffer (80 mM PIPES, pH 6.8, 1 mM MgCl₂, 5 mM EGTA) with 0.25% Triton X-100. Cells were then fixed in ice-cold methanol for 5 min before being rehydrated three times in Tris buffer (TBS, 0.15 M NaCl, 0.02 M Tris-Cl, pH 7.4) with 0.05% Tween. For immunolabeling, we used a monoclonal anti-α-Tubulin antibody produced in mouse (~2 mg mL⁻¹, monoclonal B-5-1-2, #T6074, Sigma-Aldrich). Fixed and permeabilized cells were incubated for 30 min at room temperature in a TBS-based blocking buffer (2% donkey serum, 1% BSA) and then incubated overnight at 4 °C with 1/13,000 primary antibody in blocking buffer. After three washes in TBS, the donkey anti-mouse IgG conjugated to Alexa Fluor 555 (A-31570, ThermoFisher Scientific) was applied at dilution 1/1000 in blocking buffer for 1 h at room temperature. After an extensive wash with TBS, coverslips were mounted on a glass slide with Prolong gold.

Live synaptotagmin 1 labeling: We performed labeling of synaptotagmin 1 with a mouse monoclonal purified IgG conjugated with Chromeo 488 (105 311 CR1, Synaptic Systems) targeting the lumenal domain of synaptotagmin 1. The IgGs were added to the culture medium of cells (0.3 μg mL⁻¹) for 2–4 h, after which cells were washed in PBS and fixed in PBS with 4% PFA and 40 mg mL⁻¹ sucrose. Cells were not permeabilized so that the labeling was specific to the pool of SVs that recycled by spontaneous activity during the incubation period. The labeling protocol was similar to the one with the biotinylated IgG used for QD loading (IgGs also had the same origin) for sets of labeled SVs to be equivalent in the two experiments. After three PBS washes, cells were mounted on a glass slide with Prolong gold.

Confocal imaging of immunolabeled neurons: Imaging of fluorescently labeled IgGs and of fluorescent phalloidin was performed using a commercial Zeiss confocal microscopy setup (LSM 800, ×63 plan apochromat, NA 1.4 oil objective).

**Image data analysis**. See Supplementary Methods for an exhaustive description of all data analysis methods used.

Briefly, for a kymograph $K$ the auto-correlation function at time interval $\Delta t$ was computed as the median of Pearson's correlation value for all pairs of kymograph lines that are $\Delta t$ apart (median of $\rho(K(t), K(t + \Delta t))$).

For FM 1–43 and VAMP2:mCherry images, AT-based measures of mobility were defined as

$$\text{Mobility(s}^{-1}) = \frac{\text{Cumulated distance travelled by AT}}{\text{Total duration of observation} \times \text{length of monitored axonal segments}},$$

$$\text{AT rate } (\mu\text{m}^{-1}\text{s}^{-1}) = \frac{\text{Total number of AT events}}{\text{Total duration of observation} \times \text{length of monitored axonal segments}}.$$

For Syt1-IgG:QD images they were as

$$\text{Mobility } (\mu\text{ms}^{-1}) = \frac{\text{Cumulated distance travelled by AT}}{\text{Total duration of observation} \times \text{number of QD clusters}},$$

$$\text{AT rate (s}^{-1}) = \frac{\text{Total number of AT events}}{\text{Total duration of observation} \times \text{number of QD clusters}}.$$

Actin polymerization was quantified from Utr-CH:GFP and Lifeact:tdTomato images as

$$\text{Actin polymerization } (\text{s}^{-1}) = \frac{\text{Cumulated length of new actin filament polymerization}}{\text{Total duration of observation} \times \text{length of monitored axonal segments}},$$

$$\text{Filament count } (\mu\text{m}^{-1}\text{s}^{-1}) = \frac{\text{Number of new actin filaments}}{\text{Total duration of observation} \times \text{length of monitored axonal segments}}.$$

**Statistical information**. Statistical tests were carried out as described in the main text and in figure legends. In summary, in order to compare two independent sets of measures we used the two-sided Wilcoxon rank-sum (RS) test that has the crucial advantage of being non-parametric. For paired pharmacological interventions, we used a one-sided sign (S) tests on the differences between pre and post-treatment measures. The S test is also non-parametric. The side of the tests was chosen according to prior to the manipulation effect. To test the changes in unitary properties of AT events or dynamic actin filaments, we used a multivariate analysis of variance (mANOVA) including velocity, length, and duration, as those measures were co-dependent. Multiple tests between multiple groups of univariate measures were performed with the non-parametric Kruskal–Wallis procedure. The significance values of Pearson's correlation coefficients were computed by $t$-test. We computed $p$-values for the observed number of AT events per SV trajectory (Fig. 4e) by generating 106 trials in which the same number of AT events as observed in our data were uniformly distributed at random among the same number of SV trajectories. The actual measurements of the simulated number of AT events per trajectory were compared to the empirical distribution obtained by the simulation to compute their $p$-value under the Poisson model.

**Reporting summary**. Further information on research design is available in the Nature Research Reporting Summary linked to this article.

## Data availability

The data that support the findings of this study are available from the corresponding author upon reasonable request.

## Code availability

The codes that support the findings of this study are available from the corresponding author upon reasonable request.

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

## Author contributions

N.C. and R.W.T. conceived the project. X.F. contributed some Utr-CH:GFP, Syt1-IgG: QD, FM 1-43, and VAMP2:mCherry imaging data for Figs. 1, 5, 6 and 7. N.C. performed all the other experiments and all analyses. N.C. and R.W.T. wrote the manuscript.

## Competing interests

The authors declare no competing interests.
