## [Peer Review File · Nature Communications]

Reviewers' comments:

Reviewer #1 (Remarks to the Author):

This study looks at the mobility of synaptic vesicles in hippocampal axons using a variety of techniques including FM1-43, expression of VAMP-2mcherry and the use of anti-synaptotagmin intravesicular epitope coupled to quantal dots. The authors demonstrate that nocodazole treatment selectively affects VAMP-2-mCherry but not the mobility of the internalized styryl dye or that of the QDs. They discovered that such labelled recycling vesicles are controlled by actin-based movements and free diffusion. They also demonstrate that SVs' transport is dependent on protein kinase A and actin polymerization.

I have serious conceptual and methodological issues with this study:

1-The claim that they are following recycling vesicles is not substantiated. FM1-43 is not diffraction limited and should therefore be excluded from the analysis. The authors use 2-4h incubation with QDs without a stimulatory pulse. This is highly unlikely to label recycling synaptic vesicles. A proper correlative EM analysis should be carried out to reveal the presence of or absence of QD in recycling synaptic vesicles. After 2-4 h most of the QDs are probably packaged in autophagosomes that have been found by the group of Holzbaur, to undergo retrograde transport. Acid wash should also be used to exclude the possibility that the QDs bind to the surface pool of synaptotagmin1 (see work from Mike Cousin's lab).

2- I am deeply concerned by the number of immobile carriers present in these axons. It looks like more than 50% are immobile which suggest that neurons are not healthy. Some of the figures are of very poor overall quality (Fig. 3).

3- Location of axonal segments versus nerve terminals should be included to assess the mobility of recycling vesicles in these two compartments.

4- The ability of the QDs vesicles to recycle should be assessed using re-stimulation protocol as carried out previously by this laboratory.

5- The mechanism by which the labelled structures are transported by actin is not clearly defined. The use of Nocodazole is not sufficient to demonstrate that microtubules are not involved especially if the transport is mainly retrograde (this should be determined). Some microtubules are resistant to nocodazole treatment and require low temperature change to be affected. Ad hoc methods interfering with dynein/kinesin/myosin functions should be carried out to pinpoint the precise molecular motor(s) involved in these short-range transports.

6- The mobility of recycling vesicles assessed by single molecule imaging has been characterized by the groups of Meunier and Klyachko. The latter group has actually already demonstrated that recycling vesicles mobility was controlled by acto-myosin2 network. Unfortunately, none of these studies are cited/discussed.

The nature of these major concerns precludes me from being more positive.

Reviewer #2 (Remarks to the Author):

The paper of Chenouard et al., reports on the finding that axonal traffic of recycling synaptic vesicles (SVs) between active zones relies on actin and provide some clues on how this traffic may be regulated. They also provide evidence that vesicles labeled by VAMP2:mCherry are transported by microtubules. By performing time-lapse microscopy in cultured hippocampal neurons they find that long-distance translocation of the recycling vesicles (labeled e.g., with FM dye, syt 1 Abs, or quantum dots, QDs) arise when successive bouts of active transport are linked by periods of free diffusion. The availability of SVs for active transport is promptly increased by protein kinase A and impeded by shutting off axonal actin polymerization, mediated by nitric oxide- cyclic GMP signaling leading to inhibition of RhoA.

Although some of the obtained results are improving our knowledge on the mechanisms underlying axonal transport of vesicles, the key findings of the manuscript are not completely novel. The functional significance for synaptic transmission of the regulation of the actin-dependent transport by protein kinase A and nitric oxide- cyclic GMP signaling that is uncovered in the current work still remains suggestive and requires verifications *in vivo*. PKA, for example, has many downstream actions and only a few have been followed. Therefore it is doubtful that this paper will be of interest for a broad readership of Nature Communications. In the present form the paper is more suitable for a specialized journal.

Specific comments:

1. The authors miss several earlier reports revealing that vesicle transport in axoplasm is actin-dependent (e.g., Kuznetsov et al., *Nature*. (1992) 356:722-5; Langford et al., *J Cell Sci.* (1994) 107: 2291-8; Schuh, *Nature Cell Biology* (2011) 13: 1431-6.
2. No clear explanation for differential motility of FM- and QD- tagged vesicles is provided.
3. It remains uncertain are the labeled recycled vesicles represent SVs or some larger structures. QD labeling does not allow distinguishing between those.
4. The authors state that activation of PKA by 10 μ M of forskolin (which is a rather high treatment) resulted in elongation of the presynaptic cluster and that this matched the effect of synapsin 1 deletion. However, they do not confirm this in their own experiments.
5. It has to be proven (556-558) that a reduction in actin polymerization affects presynaptic vesicle clustering. There is no evidence that supports this statement.
6. 88, e.g.,
7. 195, labeled.

Reviewer #3 (Remarks to the Author):

In this manuscript, Chenouard and colleagues studied the mechanisms of synaptic vesicle superpool's mobility using cultured neurons as a system. Using elegant labeling strategies, they found that active transport of the superpool was not dependent on MTs but instead relied on actin

dynamics. They further showed that the initiation of vesicle movements correlates with dynamic actin events. Using pharmacological tools, they showed that PKA and nitric oxide modulated the vesicle movement by affecting axonal actin dynamics.

The experiments were well designed and the analyses were very comprehensive. The findings are quite surprising and interesting. I think this manuscript should be published in Nature Communications. I just have one suggestion and one point that needs to be clarified.

Clarification: In the example provided in Fig. 3, if the time is as the arrow indicated, the examples of actin dynamic would be depolymerization events, not polymerization events. The Fig. 3B actually includes events that represent polymerization events but those were not associated with vesicle movements. Did I read this wrong?

Suggestion: in a recent paper published by the Roy lab, they reported F-actin dynamics in axons very similar to what was reported in this current manuscript. They reported a bias of actin flow in the anterograde direction. If the authors pool all the events in their control experiments, it would be interesting to see if there is an anterograde bias for the superpool movements. This might provide some hints on what the superpool might be doing.

Minor comments:

The y-axis of Fig. 2B and D should be clearly labeled.

In Fig. 2C, it is quite difficult to see the diagonal increase of Utr-CH::GFP fluorescence. Some photobleaching experiment might have the polymerization events easier to catch.

Italic: Reviewers comments. Black: the authors' response. Gray: items of factual contention. We reply first to Reviewer #3 because the answer to their main question is a prerequisite to our reply to reviewer #1.

Reply to Reviewer #3

In this manuscript, Chenouard and colleagues studied the mechanisms of synaptic vesicle superpool's mobility using culture neurons as a system. Using elegant labeling strategies, they found that active transport of the superpool was not dependent on MTs but instead relied on actin dynamics. They further showed that the initiation of vesicle movements correlates with dynamic actin events. Using pharmacological tools, they showed that PKA and nitric oxide modulated the vesicle movement by affecting axonal actin dynamics.

The experiments were well designed and the analyses were very comprehensive. The findings are quite surprising and interesting. I think this manuscript should be published in Nature Communications.

We appreciate the highly positive response and the insightful comments that followed.

I just have one suggestion and one point that needs to be clarified. Clarification: In the example provided in Fig. 3, if the time is as the arrow indicated, the examples of actin dynamic would be depolymerization events, not polymerization events. The Fig. 3B actually includes events that represent polymerization events but those were not associated with vesicle movements. Did I read this wrong?

The axes in Fig 3 were correctly labeled. However, thanks to the comment we have realized that the kymographic representation of actin polymerization was not communicated optimally and might be hard to decipher for the first-time reader. To avoid any future miscommunication, we have included a schematic rendering (revised Fig 3A) to clarify the expected patterns for interrelated F-actin polymerization and synaptic vesicle motion.

Suggestion: in a recent paper published by the Roy lab, they reported F-actin dynamics in axons very similar to what was reported in this current manuscript. They reported a bias of actin flow in the anterograde direction. If the authors pool all the events in their control experiments, it would be interesting to see if there is an anterograde bias for the superpool movements. This might provide some hints on what the superpool might be doing.

Bidirectional transport of synaptic vesicles (SVs) along axons would be functionally necessary for redistribution of presynaptic weights and was consistently seen in our experiments (Fig 1E, 1I, 2A; Rebuttal Fig 1) and in earlier literature (Krueger, Kolar, and Fitzsimonds, Neuron 2003). To move beyond qualitative observations regarding possible directional bias in SV active transport (AT), we have now performed additional trajectory analysis (new Fig. S11, also included here as Rebuttal Fig 1. Our analysis of 327 Syt1-IgG:QD trajectories displaying multiple AT events showed that the distance travelled by SVs could be consistently and parsimoniously explained by an unbiased random-walk model, with equal probability of motion and equal AT distance in each direction. We thank the reviewer for courteously encouraging the new analysis.

At first glance, our finding would appear to be at variance with the ~25% difference in average numbers of F-actin trails between anterograde and retrograde directions found by the Roy lab (N=28, number of observation periods, not the number of individual events) (Ganguly et al., J Cell Biol, 2015). We have commented on that matter in the text [new L 44-45, 597] and considered various possible explanations.

First, experimental procedures could differ between labs and imaging protocols. For instance, with vesicular markers (FM dyes, QDs) proximal, medial and distal parts of axons are imaged without bias as the many segments in the field of view are invisible. This balance might differ for Utr-CH:GFP imaging when the experimenter chooses axonal segments, likely based on other criteria. We have noticed considerable variability in F-actin polymerization along the heterogeneous, long axons depending upon proximity to the initial segment or growth cone. Second, because vesicle transport is not directly driven by actin polymerization, but merely upper-bounded by it, there is no overwhelming reason to expect a precise 1:1 relationship between actin polymerization and SV AT; other factors may come into play. Third, to restate briefly our main point, we can firmly refute a consistent and strong directional bias in SV transport as would be expected and was in fact seen with microtubule-based transport. Indeed, we confirmed this by applying the same analysis to VAMP2:mCherry transport (Fig S11 and Rebuttal Fig 1, which we have shown to rely on microtubules (Fig 1); this provided a good control of the sensitivity of the method. Thus, SV traffic is unlike the transport of many other organelles which has been shown to be strongly directionally biased and to rely on microtubule-based transport.

We are grateful to the Reviewer for uncovering this interesting matter which we will investigate further and could yield biological insight.

Minor comments:

The y-axis of Fig. 2B and D should be clearly labeled.

We now do so in the revised Fig 2.

In Fig. 2C, it is quite difficult to see the diagonal increase of Utr-CH::GFP fluorescence. Some photobleaching experiment might has the polymerization events easier to catch.

In order to improve the visualization of F-actin polymerization events corresponding to the kymographs in Fig 2C we have included a new analysis in Revised Supplementary Figure 9.

As rightfully pointed out by the Reviewer, obtaining crisp images of dynamic actin in alive neurons without interfering with the actin physiology is known to be difficult (Ladt et al. Methods in Cell Biology 2016). When using Utr-CH:GFP as a probe, low contrast stems from the low concentration of the fluorescent molecule and the background signals from the free (F-actin-free) probe. Unfortunately, low Utr-CH:GFP expression is required to safeguard normal actin physiology (revised Fig S8) (Ganguly et al., J Cell Biol, 2015). Moreover, free Utr-CH:GFP (the unwanted background) cannot be easily bleached without signal degradation because its aggregation as fluorescent probe on newly formed F-actin is precisely what allows elongating filaments to be detected.

We implemented additional image processing to digitally subtract the background Utr-CH:GFP fluorescence, in the same spirit as the bleaching experiments suggested by the Reviewer. The local fluorescence is integrated over a 5 s time-window, just prior to the time point of interest and subtracted from the instantaneous fluorescence signal, thus discarding local, slowly-varying, fluorescence signals arising from free probe molecules and static structures. It will be evident that polymerization events identified in Fig 2C are clearly enhanced in revised Fig S9D; the lack of polymerization after a brief treatment with latrunculin A will also be more obvious. The same approach was used to enhance Utr-CH:GFP images in revised Fig 3A and C. At the beginning of actin part, we opted to stay close to

unprocessed data, keeping raw Utr-CH:GFP images in the main text figures (Fig 2), while introducing the usefulness of the digitally subtracted images in revised Fig S9D.

We thank the Reviewer for the perceptive comments which prompted a substantial improvement of the work.

Reply to Reviewer #1

This study looks at the mobility of synaptic vesicles in hippocampal axons using a variety of techniques including FM1-43, expression of VAMP-2mcherry and the use of anti-synaptotagmin intravesicular epitope coupled to quantal dots. The authors demonstrate that nocodazole treatment selectively affects VAMP-2-mCherry but not the mobility of the internalized styryl dye or that of the QDs. They discovered that such labelled recycling vesicles are controlled by actin-based movements and free diffusion. They also demonstrate that SVs' transport is dependent on protein kinase A and actin polymerization.

We thank the reviewer for this compact, factual summary.

I have serious conceptual and methodological issues with this study:

1-The claim that they are following recycling vesicles is not substantiated. FM1-43 is not diffraction limited and should therefore be excluded from the analysis [addressed separately below]. The authors use 2-4h incubation with QDs without a stimulatory pulse. This is highly unlikely to label recycling synaptic vesicles.

The second, more serious comment stems from a fundamental miscommunication. Our neurons were incubated with Syt1-IgG:QDs in high K⁺ solution for only 90 s, not 2-4 h at rest. This standard hyperkalemic pulse is known to trigger SV exocytosis and recycling (e.g. Pyle et al., Neuron 2000) (Rebuttal Fig 2) and its brevity minimizes non-evoked endocytosis of non-SV organelles. Thus, we did use a stimulatory pulse and kept the exposure to quantum dots (QDs) brief. This was clearly specified in the main body of the manuscript [quotes and (old) and [new] locations of items in question are listed below in gray for the convenience of the reviewers and editor].

(L 183-184)[L 319] “To render such vesicles brightly fluorescent, hippocampal neurons were further stimulated with **45 mM K⁺ for 90 s** in the presence of streptavidin-coated QDs.”

(L 836-837)[L 743] and in the Methods section: “Streptavidin-coated QDs with 655 nm peak emission (ThermoFisher Scientific, Cat. No. Q10123MP) were applied in a **high potassium Tyrode's solution containing 2% BSA for 90 s.**”

Here is the likely source of confusion: before the 90-s stimulus for QD uptake, we preincubated neurons for 2-4 h, not with QD themselves, but with a biotinylated antibody against synaptotagmin 1. This extra step was designed to increase the likelihood of labeling syt-1-containing synaptic vesicles. A classical paper from De Camilli pioneered uptake of anti-syt-1 Ab as a way to label recycling SVs (Kraszewski et al., J Neuro 1995) (L 182) [new L 164]. The reviewer would be correct in pointing out that some antibody winds up in other compartments than SVs during the prolonged incubation, but this doesn't matter because the exposure to streptavidin-coated QDs occurs only during the 90 s pulse of 45 mM K⁺, ensuring that only Syt1-IgGs in recycling SVs were secondarily labeled with QDs.

A proper correlative EM analysis should be carried out to reveal the presence of or absence of QD in recycling synaptic vesicles. After 2-4 h most of the QDs are probably packaged in autophagosomes that have been found by the group of Holzbaur, to undergo retrograde transport.

If we put ourselves in the reviewer's shoes, thinking that the QDs were loaded over 2-4 hr, this question makes perfect sense. In any case, Reviewer #1 is right to point that electron microscopy (EM) is a powerful tool to validate the presence of QDs inside SVs. This technique has been reliably used to validate the SV localization of intravesicular probes such as of nanobodies targeting VAMP2 (Joensuu et al., J Cell Biol,

2016); in the same vein, our lab has already published EM images confirming that SVs take up QDs in a 1:1 fashion (Zhang, Cao and Tsien, PNAS 2007) (Rebuttal Fig 3).

(L 179-180) We had already pointed to this study in the manuscript: “labeling single vesicles with quantum dots (QDs), which are bright, resistant to photobleaching, and taken up by SVs in a 1:1 fashion [as verified by EM][22]” but we have stated this more clearly now [addition in brackets].

To follow up with the Reviewer’s comment we have further highlighted the important information of EM microscopy validation of SVs in additions to the revised manuscript [new L138-141, L 161].

“Acid wash should also be used to exclude the possibility that the QDs bind to the surface pool of synaptotagmin1 (see work from Mike Cousin’s lab).”

To the best of our knowledge, Mike Cousin has not used QD-based techniques to label SVs. He has used dextran-based probes (Clayton and Cousin, J Neurosc Methods 2009), FM dyes (Clayton et al., Nature Neurosc 2010) and lately has made extensive use of pHluorin tags (Nicholson-Fish et al., Neuron 2015; Zhang et al., J Neuro 2015). Nonetheless, we thank the reviewer for pointing us to Cousin’s interesting work.

Apologies in advance for highlighting matters the reviewer is familiar with, but this will help reviewer, editor and author all to be on the same page. External acidification makes sense for pHluorins as they are pH-sensitive fluorescent probes (Miesenböck et al., Nature 1998) – it quenches them. Acidification is not very effective in quenching QDs fluorescence (QDs are rather pH-insensitive). Thus, we interpret the reviewer’s statement as a general concern about signals from inappropriately non-internalized QDs. Here our use of chemical quenchers potentially achieved the same objective as we think the reviewer meant. First, non-specific binding of QDs was mitigated by inclusion of bovine serum albumin during the 90-s incubation with Syt1-IgG:QDs. Second, and most important, all the fluorescence data was gathered with a fluorescence quencher in the extracellular solution, thus rendering invisible any QD left on the external surface (that is, not tagging SVs).

(L 804-806) This was clearly stated: “For QD imaging, 1 μ M BHQ-3 (BHQ-3001-5, Biosearch Technologies, Petaluma, CA) or 5 μ M trypan blue (Sigma-Aldrich) were used to quench the fluorescence of extracellular QDs”.

Further, we carefully titrated the chemical compounds for maximal efficiency and minimal toxicity as shown with experimental quenching curves presented in Rebuttal Fig 2A. Also, we systematically imaged QD fluorescence in our cells before and after addition of the external fluorescence quencher to check the efficiency of the quenching protocol (Rebuttal Fig 2B).

But, taking the reviewer’s underlying concern to heart, we have clearly indicated in the revision the use of a fluorescence quencher in the main body of the text [new L 171] and included Rebuttal Fig 2 as a new Supplementary Figure (revised Fig S6) showing the high quenching potency for external QDs. Any future confusion or miscommunication has been forestalled in the revision by including further corroboration of the high level of specificity of our protocols for labeling recycling SVs with QDs [new L 169-174].

“FM1-43 is not diffraction limited and should therefore be excluded from the analysis.”

The reviewer seems to trust QD measurements, as we do for specific purposes, but we contend that for a general audience a second, highly familiar method will be highly reassuring, even if the optical properties

are less favorable than for QDs. Here we spell out why FM dye experiments are useful even if not used to track single vesicles, and why their labeling of vesicles is specific to recycling vesicles.

Several lines of evidence show that SVs often aggregate in motile multibody clusters for transport (Krueger, Kolar, and Fitzsimonds, *Neuron* 2003; Darcy et al., *Nat Neurosc* 2006; Staras et al., *Neuron* 2010; Gramlich et al., *Cell Rep* 2017) which, as rightly pointed out by the Reviewer, may not be diffraction-limited. For this reason, we repeatedly used the term ‘cluster’ to indicate this point clearly throughout the manuscript (e.g. L 163) [new L 144, 150]. For example, in Fig 1 we measured the dynamics of SV clusters’ center of mass, not the position of individual SVs. Tracking the cluster’s center of mass is a valuable index of SV population dynamics though it cannot replace the tracking single SVs that we also performed (Fig 4). Results based on the dynamics of FM-dye-labeled clusters have been repeatedly published (Krueger, Kolar, and Fitzsimonds, *Neuron* 2003; Darcy et al., *Nat Neurosc* 2006; Staras et al., *Neuron* 2010). For studies of transport mechanisms, it enables collection of large data sets (e.g. N=4231 active transport events in Fig 1G-H) for robust statistical analysis. This avoid missing an effect because of insufficient data, a pitfall the Reviewer would surely want us to avoid.

Though the reviewer’s concern about QD targeting was partly based on a misunderstanding, we anticipate further questions about the specificity of FM dye targeting as well. To demonstrate directly that the FM-filled vesicles do destain, and hence are recycling vesicles, (Fig 1), we have consistently measured FM dye destaining – corresponding to SV exocytosis – in a response to mild trains of electrical stimuli (100APs 10Hz, original manuscript Fig 4C) or after application of high K⁺ solution (Rebuttal Fig 4). These procedures are well-accepted demonstrations of the recycling and fusion ability of the stained vesicles, and were performed routinely here. Photoconversion+EM experiments by us and by others clearly show that FM dye loading stains morphologically identifiable synaptic vesicles (e.g. Harata et al., *PNAS* 2001; Rizzoli and Betz, *Science* 2004; Gaffield and Betz, *Nat Protoc*, 2006) (Rebuttal Fig 5). To reassure Reviewer #1 and readers, the revised manuscript includes additional data from FM-dye destaining experiments (Rebuttal Fig 4 included as revised Fig S5) [new L 141-142].

While we agree FM dyes can marginally label other compartments, the literature is unambiguous that the vast majority of stained organelles are bona fide recycling vesicles (e.g. Schikorski and Stevens, *Science*, 2001; Harata et al., *PNAS* 2001; Rizzoli and Betz, *Science* 2004; Gaffield and Betz, *Nat Protoc*, 2006). Marginal staining of non-SV compartments could not plausibly explain the lack of effect of nocodazole demonstrated in Fig 1E-H.

If Reviewer #1 remains dissatisfied with inclusion of the dynamics of FM-dye-labeled SVs, we are willing to compromise, giving them less prominence by moving FM1-43 dye-based results out of Fig 1 and to a Supplementary Figure to join the aforementioned extra validation results (Rebuttal Fig 4). We believe that many in the field would want to see that results of QD-labeling are largely mirrored by FM-based experiments, even if their spatial resolution is far less.

“2- I am deeply concerned by the number of immobile carriers present in these axons. It looks like more than 50% are immobile which suggest that neurons are not healthy.”

We respectfully disagree: rigorous scientific studies have unequivocally shown that a vast majority of SVs are confined to the synaptic boutons. Thus, finding a majority of carriers showing little mobility is to be expected and is not a sign of bad health. Yukiko Goda quantified the proportion of extrasynaptically mobile SVs as ~20% of the total synaptic pool over ~20 min of observation (also confirmed in Gramlich

and Klyachko, Cell Rep 2017), in good agreement with our findings [new L 336]. Inside the synapse, our mobility measures matched that of the literature and we have clearly indicated this good fit in the revised manuscript by including appropriate references to avoid confusion [new L 369, 371, 574-575]. Indeed, SVs were not strictly “immobile”: we have found on average a sub-diffusive mode of motion (Fig 4F) which agreed with structural data showing SV tethering (Hirokawa et al., J Cell Biol, 1989). The distance travelled by SVs inside the synapse saturated after 4 s at ~100 nm on average (Fig 4F), an excellent match with previous dynamical data showing a confinement cage of radius ~50-150 nm (Lemke and Klingauf, J Neurosci 2005; Jordan et al., Biophys J, 2005; Peng et al., Neuron 2012; Forte et al., J Neurosci 2017). Our average population analysis did not emphasize SVs with time-varying diffusion coefficients and modes as it is beyond the scope of this study; other works, which we now cite [L 574-575], have precisely quantified it using powerful statistical methods such as hidden Markov models (Joensuu et al., J Cell Biol, 2016, Forte et al., J Neurosci 2017).

SV membrane fusion triggered by hyperkalemic and electrical stimuli, as judged by FM dye and QD destaining (Fig 4, Rebuttal Fig 2 and 4), are other signs of fully functional synapses and good health. Extra evidence of SV fusion-competence have therefore been included in the revision (Fig S5, S6) [L 138-142, 171-174].

“Some of the figures are of very poor overall quality (Fig. 3).”

Stimulated by this remark we have enhanced Fig 3 and included results from a new visualization technique (supplementary Methods) that allows highlighting actin polymerization events in previously low-contrast Utr-CH:GFP images (Revised Fig 3 and Fig S9), hence consolidating the presented results (for details, see below). We provided an explanation of the constraints on the use of Utr-CH:GFP as an actin probe in our reply to another reviewer (Reviewer #3, see above).

For Fig 3, specifically, we first want to underscore the technical challenge of simultaneously imaging nanometer-length motion of synaptic vesicles and fast actin polymerization at sub-second scales, while avoiding overexpression of optical probes which is a major concern for Lifeact/Utr-CH-type probes (as shown in Revised Supplementary Fig 8 and by the group of S. Roy; Ganguly et al., J Cell Biol, 2015). Such simultaneous information has never been presented before, possibly reflecting the technical difficulties of working with limited optical signal strength.

To compensate for low contrast in Utr-CH:GFP images, we devised a new analysis and visualization method ($\Delta F_{\text{Utr-CH:GFP}}$, see revised Methods) that facilitated the identification of fast polymerization events despite challenging optical and biological constraints. Fast F-actin polymerization was objectively and unambiguously highlighted (revised Fig 3A, right; Fig 3C, right) by canceling out slowly varying background signals and applying an image de-noising method that preserved edge-like structures (Beck and Teboulle, IEEE Trans Image Proc, 2009). Moreover, we included a didactic schema of the idealized optical signals for coordinated AT of SVs and longitudinal F-actin formation (Revised Fig 3A, bottom). This proved to be an excellent match to our measures in live axons and clarified the data interpretation (Fig 3A, top; Fig 3C, bottom). We also included new $\Delta F_{\text{Utr-CH:GFP}}$ images in the revised Fig S9 which validated the reliability of the identification of longitudinal actin polymerization based on Utr-CH:GFP fluorescence: events identified in Utr-CH:GFP images yielded crisp diagonal patterns in $\Delta F_{\text{Utr-CH:GFP}}$ images which were absent after disruption of actin dynamics (revised Fig S9D).

To improve the quality of Fig 3 we have also enhanced panel D and provided a clarified explanation of our results [L 294-299].

We thank the Reviewer for prompting a significant improvement of the quality and clarity of Fig 3 despite strict technical constraints. Their comment was also valuable in encouraging us to consolidate the Utr-CH:GFP-based results and implement further validation of the approach.

“3- Location of axonal segments versus nerve terminals should be included to assess the mobility of recycling vesicles in these two compartments.”

The synaptic/extrasynaptic localization of SVs is indeed important information. This point has already been addressed in Fig 4 where we delineate recycling vesicle clusters using maps of FM dye destaining during electrical stimulation. We have found no occurrence of active transport within synapses themselves (Fig 4D). Hence, the active transport-based motility studies described in the other figures are specific to extrasynaptic compartments. We have re-emphasized this point in the revised manuscript to avoid any possible confusion [L 572-574].

Regarding intrasyaptic vesicle movements, we point out that there is already a vast body of work on the underlying mechanisms (to name only a few: Krazsewski et al., J Neurosci 1996; Li and Murthy, Neuron 2001; Shtrahman et al., Biophys J 2005; Gaffield, Rizzoli and Betz, Neuron 2006; Jordan, Lemke and Klingauf, Biophys J 2006 Westphal et al., Science 2008; Kamin et al., Biophys J 2010; Peng et al., Neuron 2012; Joensuu et al., J Cell Biol, 2016; Forte et al., J Neurosci, 2017). Indeed, our observation of a predominance of constrained diffusion inside synaptic clusters is in good agreement with the literature (in the revised manuscript, we now provide appropriate bibliographical references to the topic of intrasynaptic mobility [L 46, 369, 371, 574-577]). Experimentally, however, our main goal was to focus on more uncharted territory, the mechanisms of long-range, extrasynaptic mobility. Our work extends and is highly complementary to recent work on extrasynaptic traffic and longitudinal actin dynamics (Ganguly et al., J Cell Biol, 2015), myosin V support of SV transport (Gramlich and Klyachko, J Cell Sci 2017), and the control of axonal traffic of large organelles by contractile actin rings (Wang et al., bioRxiv 492959, 2018).

“4- The ability of the QDs vesicles to recycle should be assessed using re-stimulation protocol as carried out previously by this laboratory.”

We concur with the reviewer that this could be a useful piece of information for the reader and thank him/her for this suggestion. We were able to trigger the exocytosis of the content of QD-containing SVs upon electrical stimulation with a train of pulses (Rebuttal Fig 4D), and even time-locked to single pulses (Rebuttal Fig 4C). Recycling-ability is implied by the QD-staining protocol, which relies on endocytosis. In deference to the reviewer's suggestion, we now cite past studies that used QD-based tagging of synaptic vesicles and showed their membrane-fusion competence (Zhang, Cao and Tsien, PNAS 2007; Zhang, Li and Tsien, Science 2009; Park, Li and Tsien, Science 2012; Lee et al., PlosOne 2012) [L 171-174]. To make the fusion-recycling competence of SV-QDs explicit, we now include Rebuttal Fig 4 as revised Fig S6.

“5- The mechanism by which the labelled structures are transported by actin is not clearly defined. The use of Nocodazole is not sufficient to demonstrate that microtubules are not involved especially if the transport is mainly retrograde (this should be determined). Some microtubules are resistant to nocodazole treatment and require low temperature change to be affected.”

We thank the reviewer for the constructive comment. Our answer has three facets.

(A) We stand by our experiments with nocodazole. Previously, nocodazole has been used to interrupt the traffic of synapsin:PAGFP proteins (Scott et al., Neuron, 2011) and VAMP2:GFP (Song et al., Cell 2009) in axons of cultured hippocampal neurons, hence demonstrating its potency. It has also been routinely used in other systems to disprove a role for microtubules in organelle traffic (eg Schuh, Nat Neurosc 2011). We did not further elevate the severity of the nocodazole treatment to avoid imperiling cell health and axonal structures in a manner that would have confounded any measure of traffic. Two experiments strongly pointed to the effectiveness of our nocodazole treatment and successful depolymerization of a significant fraction of microtubules: (1) immunocytochemistry for α -tubulin showed defects of the cell cytoskeleton (Fig S2). (2) Nocodazole very potently interrupted the traffic of large VAMP2:mCherry clusters (Fig 1A)– a positive control – which are known to rely on microtubules for traffic (Song et al., Cell, 2009; Hirokawa, Niwa and Tanaka, Neuron 2010; Nirschl, Ghiretti and Holzbaaur, Nat Rev Neurosc 2017).

(B) Still, as correctly indicated by the Reviewer, we cannot totally exclude that a small, very resistant, fraction of microtubules supports SV traffic (but not motion of VAMP2 or synapsin packets) in coordination with actin filaments. We now consider this intricate scenario in the discussion section of the revised manuscript [L 534-537] and present nocodazole-based results in a more conservative manner [L 80, 130-132, 156-157].

(C) We now provide a new supplementary figure (Fig S16, see also Rebuttal Fig 6) [L 534-537] that presents results obtained with colchicine, a microtubule-depolymerizing drug with a different mechanism of action than nocodazole (Skoufias and Wilson, Biochemistry, 1992). It showed no negative effect on SV traffic, in excellent agreement with the nocodazole-based experiments (Fig 1).

(D) We have now elaborated on the issue of directionality, both in the revised text [L 384-393, 541] and in the Reply to Reviewer #3, repeated here for convenience.

Qualitatively, both the literature (Krueger, Kolar, and Fitzsimonds, Neuron 2003) and our experiments (Fig 1E, 1I, 2A; Rebuttal Fig 1) have consistently shown bidirectional transport of SVs along axons, as expected if synaptic weights were to be redistributed, in part presynaptically. To quantify a potential directional bias in SV active transport (AT), we have performed extra trajectory analysis shown in a new supplementary figure (revised Fig S11, included as Rebuttal Fig 1). The analysis of 327 trajectories of Syt1-IgG:QD displaying multiple AT events consistently showed that the distance travelled by SVs could be parsimoniously explained by an unbiased random-walk model, with equal probability of motion and equal AT distance in each direction. This preliminary data therefore firmly refutes the hypothesis of a consistent and strong directional bias in SV transport. On the contrary, VAMP2:mCherry transport, which we have shown to rely on microtubules (Fig 1), was strongly biased when we applied the same analysis (Rebuttal Fig 1), in good agreement with the presumption of the Reviewer that microtubule-based transport is directionally biased. SV traffic is therefore unlike the transport of many other organelles which relies on microtubule-based transport and has been shown to be directionally biased. In the revised manuscript, this new analysis is now included in Fig S11, along with discussion of this interesting feature of SV traffic and its contrast to other, more conventional, biased traffic along axons [L 384-393, 544]. We thank the reviewer for pointing our attention to this important issue and prompting the new analysis.

“Ad hoc methods interfering with dynein/kinesin/myosin functions should be carried out to pinpoint the precise molecular motor(s) involved in these short-range transports.”

We agree with the Reviewer that our submission did not provide direct evidence for the detailed roles of motor proteins from different families. Spurred by the Reviewer's comment and the demonstrated roles of myosin II in intrasynaptic mobility (reviewed in Li et al., Trends in Neurosc 2019), we have now explored the possibility that myosin II plays a role in regulating extrasynaptic traffic in a new set of experiments. Traffic of SVs labeled with Syt1-IgG:QD was monitored before and after treatment with ML-7, a myosin light chain kinase inhibitor (Saitoh et al., J Biol Chem, 1987) which potently inactivated myosin II in synapses (Ryan, J Neurosci, 1999; Jordan et al., Biophys J, 2005; Peng et al., Neuron 2012) and at the axon initial segment (Berger et al., Neuron 2018). Treatment for 15 min with 20 μ M ML-7 failed to elicit a detectable decrease of SV traffic (median motility change +7.0%; $p=1$, two-sided sign-test; N=6) (rebuttal Fig 7), hence arguing against a major role of the acto-myosin2 complex in SV transport outside of synapses. This new set of results has been included in the revised manuscript as a supplementary figure (Fig S17) [L 575-578].

Compelling evidence for a key role of myosin V has already been published by Klyachko's group (Gramlich and Klyachko, Cell Rep 2017) and is consistent with our own findings of repetitive AT along long actin filaments (Fig 3) and lack of major role for myosin II. Repeating the same experiments as Klyachko's will not provide original results, greatly advance knowledge, or increase the novelty of our study. Likewise, given the lack of effect of microtubule depolymerization on extrasynaptic SV traffic, investigating a hypothesized role of dynein and kinesin-type motors is unlikely to bear fruit. Such an investigation is made more problematical because (1) no generic drug exists for motors of the kinesin family; (2) genetic interference with kinesin motors is complicated by the many protein variants in this family and the important side-effects of durably knocking down/off such important cell function. Nonetheless, in an effort to comply with the Reviewer's suggestion, we tested the dynein inhibitory drug Ciliobrevin D, but found it to cause excessive toxicity, even for a light treatment (concentration < IC_{50}) (Rebuttal Fig 8). We hope that the Reviewer will agree that further testing of microtubule-based motors would be a high cost/low payoff gambit for enhancing the current work on extrasynaptic SV traffic, given our observations in Fig 1.

"6- The mobility of recycling vesicles assessed by single molecule imaging has been characterized by the groups of Meunier and Klyachko. The latter group has actually already demonstrated that recycling vesicles mobility was controlled by acto-myosin2 network. Unfortunately, none of these studies are cited/discussed."

We thank the Reviewer for his/her interesting suggestion. As previously stated, the focus of this work is extrasynaptic mobility and its regulation in the context of neuronal plasticity. We have tried harder to cite fairly past findings about intrasynaptic mobility, including the Meunier and Klyachko papers mentioned (Peng et al., Neuron 2012; Joensuu et al., J Cell Biol, 2016) [L 46, 141, 369, 371, 573-577]. This is an improvement because of the striking differences between mobility mechanisms in extra- and intra-synaptic domains.

If we cell biologists step back from specific studies and consider the big picture, all this makes functional sense. Structurally, the extra-synaptic space displays dynamic, multi-micrometer length actin filaments (Xu, Zhong and Zhuang, Science 2013; Ganguly et al., J Cell Biol 2015), which we showed as a support for SV traffic (Fig 2 and 3). In contrast, inside synapses, actin has been described as a dense meshwork of short filaments ($\ll 1 \mu$ m) (Fifkova & Delay, J Cell Biol, 1982; Hirokawa et al., J Cell Biol, 1989) which may participate in controlling the access of the SVs to exocytosis sites and in regulating their retrieval (reviewed

in Li et al., Trends in Neurosc 2019). The acto-myosin2 complex has the ability to control the tensile strength of the synaptic actin meshwork (Li et al., Trends in Neurosc 2019) and of the axon radius by interaction with the circumferential actin-spectrin rings outside the synapse (Fan et al., Sci Rep, 2017; Wang et al., bioRxiv 492959, 2018). However, for longitudinal motion along axons, none of these myosin II functions seem as suitable as those of myosin V - a processive motor that directly binds SVs (Prekeris and Terrian, J Cell Biol 1997) and has been implicated in extrasynaptic SV traffic (Gramlich et al., Cell Rep 2017). Myosin V seems well adapted to transport SVs on micrometer-long journeys along the axon by following the tip of a rapidly growing actin filament (Fig 3). Our argument is given further experimental weight by the new set of paired experiments with ML-7 (discussed above) that showed no role for myosin II in active transport of small SVs outside of synapses (rebuttal Fig 7, Fig S17). We have included appropriate discussion in the revised manuscript [L 572-581].

The nature of these major concerns precludes me from being more positive.

We apologize for any lack of clarity on our part and hope that the new analysis, revised explanations, and clarification of regrettable miscommunications have improved the manuscript to the degree required by the expert reviewer.

Reply to Reviewer #2

The paper of Chenouard et al., reports on the finding that axonal traffic of recycling synaptic vesicles (SVs) between active zones relies on actin and provide some clues on how this traffic may be regulated. They also provide evidence that vesicles labeled by VAMP2:m Cherry are transported by microtubules. By performing time-lapse microscopy in cultured hippocampal neurons they find that long-distance translocation of the recycling vesicles (labeled e.g., with FM dye, syt 1 Abs, or quantum dots, QDs) arise when successive bouts of active transport are linked by periods of free diffusion. The availability of SVs for active transport is promptly increased by protein kinase A and impeded by shutting off axonal actin polymerization, mediated by nitric oxide- cyclic GMP signaling leading to inhibition of RhoA.

We appreciate the detailed summary of some of the results and implications and the playback of what concepts came across.

Although some of the obtained results are improving our knowledge on the mechanisms underlying axonal transport of vesicles, the key findings of the manuscript are not completely novel. 1. The authors miss several earlier reports revealing that vesicle transport in axoplasm is actin-dependent (e.g., Kuznetsov et al., Nature. (1992) 356:722-5; Langford et al., J Cell Sci. (1994) 107: 2291-8; Schuh, Nature Cell Biology (2011) 13: 1431–6.”

Actually, we did reference the Schuh paper in Nat Cell biology, in the Discussion; we now have happily included the other references as well.

As to the issue of novelty, we hope we can find agreement with the Reviewer that there is room for distinguishing between recycling synaptic vesicles and organelles different from SVs, and between squid axons and axons of small neurons in the mammalian CNS, whose uses for cell biological plasticity are likely to be very different. All these systems are of great interest, but there is good reason to put insights from model organisms and in vitro systems to the test in studies of neurons from circuits participating in learning and memory. We believe that confirming or (dis)proving concepts about the axonal cytoskeleton and traffic from distant preparations is necessary in the mammalian CNS. Indeed, thanks to our dual-monitoring experiment (Fig 3), we found that actin polymerization preceded SV motion, rather than stimulated it, in contrast to transport of Rab11a-containing vesicles in mouse oocytes (Schuh, Nat Neurosc 2011) [new L554-556]. We have now taken pains to acknowledge the importance of actin-, and microtubule-dependence of organelle movement as studied in extruded axoplasm from squid giant axons (Allen et al., Science 1982; Brady et al., Nature 1984; Brady et al., Cell Motility, 1985; Schnapp et al., Cell, 1985; Weiss et al., Cell Motil, 1988; Kuznetsov et al., Nature, 1992; Langford et al., J Cell Sci, 1994; Tabb et al., J Cell Sci, 1998) and giant cerebral neuron of Aplysia (Goldberg et al., PNAS 1980) [new L 39-41, 564-567]. However, for SVs in the mammalian CNS, specifically, our findings ruled out a major contribution of microtubules to traffic support. Moreover, properties of longitudinal actin filaments seem to differ between squid axoplasm and our study system: actin stabilization with phalloidin (which shares its actin-stabilizing properties with jasplakinolide) was found to promote filament formation and traffic (Goldberg et al., PNAS, 1980; Brady et al., Cell Motility, 1985; Langford et al., J Cell Sci, 1994; Tabb et al., J Cell Sci, 1998), whereas jasplakinolide exerts a clear inhibitory effect in intact axons from small CNS neurons as shown by us and by others (Darcy et al., Nat Neurosc 2006). By use of up-to-date tools, we now find transport along transiently polymerizing actin filaments that has not been reported before in axons. This in no way diminishes the importance of studies in squid axons (work pioneered by Langford, Weiss, Kuznetsov, Brady, Lasek and others), where dynamic changes in actin have not been studied and may not

be necessary. Nonetheless, the emphasis is very different. Our observations have the potential to significantly change the way we think about dynamic resource re-distribution along CNS axons, contributing to the originality and future impact of our work.

“The functional significance for synaptic transmission of the regulation of the actin-dependent transport by protein kinase A and nitric oxide- cyclic GMP signaling that is uncovered in the current work still remains suggestive and requires verifications in vivo.” “PKA, for example, has many downstream actions and only a few have been followed.”

We agree that the mechanisms that we uncovered for vesicle redistribution are suggestive. Results in Figures 5 and 6 are intended to provide first steps toward understanding the interplay between dynamic cytoskeletal remodeling, synaptic vesicle reallocation, and synaptic plasticity. We do however want to underline the complementarity with the work of the Columbia group (Wang, Kandel, Hawkins, Antonova et al.), summarized in their title: “Presynaptic and Postsynaptic Roles of NO, cGK, and RhoA in Long-Lasting Potentiation and Aggregation of Synaptic Proteins”. By performing extensive experiments to dissect the mechanisms of SV traffic (Fig 1-4), and providing mechanistic links (Figures 5 and 6) to modulation by PKA and NO, widely accepted mediators of synaptic plasticity, we now tie together studies of neuronal cell biology and synaptic plasticity. Because the bulk of our work was dedicated to understanding basic mechanisms of SV traffic, we propose to leave further detailed description of the interplay with synaptic plasticity to a follow-up study.

While the characterization of SV traffic in live animals would be ideal, it is impossible to do so with the current technological arsenal. First, the specific labeling of recycling synaptic vesicles was never done in live animals. The limited specificity of genetically expressed probes, such as VAMP2:mCherry (Fig 1), preclude their use. Second, most active transport events are sub-second and sub-micron long, hence requiring very high resolution and sensitive imaging, which is challenging to implement in live rodents. Third, targeted and rapid pharmacological manipulations are also challenging to implement in live animals; to the best of our knowledge, only Herzog and colleagues have implemented a protocol for studying SV traffic *in vivo*, supporting the concept of vesicle superpools in mice (Herzog et al., J Neurosci 2011). However, the monitoring technique was not highly sophisticated (FRAP experiment with genetic probes) and no manipulation could be performed.

We completely agree with the Reviewer that PKA plays multiple roles in neurons. We were less concerned with pinpointing the exact molecular mechanism of PKA action than with gaining a systematic categorization of how PKA might influence the extrasynaptic traffic of SVs. As suggested by the Reviewer, PKA could exert its control over SVs through different downstream pathways impacting distinct aspects of the trafficking process: 1/ the availability for active transport of SVs trapped in synaptic clusters, 2/ the availability of F-actin tracks, 3/ the efficiency of molecular motors. Our data refutes options #2 and #3, as we have found no effect of Forskolin on (#2) dynamic F-actin formation (Fig 5D and Revised Fig S12, previously Fig S8) and (#3) unitary properties of SV active transport (Revised Fig S12, previously Fig S8). Instead, our data supports option #1. We meant Fig 5 as an overall phenomenological summary of our empirically observed effects of PKA activation on SV traffic and longitudinal actin polymerization, observations that are valuable because of their broader implications for neuronal plasticity even without further exploration of the exact mechanism by which PKA recruits SVs. The more specific explanation we offer (#1), the mobilization of vesicles from their synaptic synapsin tethers, is grounded in considerable rigorous scientific work (in Paul Greengard’s and Daniel Gitler’s groups for example), and this known

liberation of SVs by PKA-mediated phosphorylation of synapsin fits nicely with our observations of synaptic cluster elongation (Fig 5A) and of increased incidence of active transport events (Fig 5B). To be sure that readers are not too fixated on the specific scenario of PKA-> synapsin phosphorylation->vesicle mobilization, we now mention in the text other avenues for vesicle mobilization that have not directly implicated PKA, in particular the disruption of cadherin- β -catenin (Bamji et al., J Cell Biol 2006) and VAMP2- α -synuclein interactions (Diao et al., Elife 2013) [new L 395-396]. We also considered PKA actions on targets other than synapsin: PKA->melanophilin phosphorylation->cargo switch to microtubule tracks (Oberhofer et al., PNAS 2017) and PKA->myosin light chain kinase (MLCK) inhibition->myosin II relaxation (Conti & Adelstein, J Biol Chem 1981), which would have fallen under options #2 or #3 had these received experimental support. Indeed, melanophilin transcripts are not found in principal cells of the hippocampus (HIPPOSEQ search, <https://hipposeq.janelia.org>) and traffic of SVs labeled with Syt1-IgG:QD was insensitive to inhibition of MLCK with ML-7 (Saitoh et al., J Biol Chem, 1987) (rebuttal Fig 7, included as revised Fig S17). We now briefly discuss the hypothesis of PKA/myosin II- and PKA/myosin V-mediated regulation of SV traffic (new L 575-578, 608-609).

“Therefore it is doubtful that this paper will be of interest for a broad readership of Nature Communications. In the present form the paper is more suitable for a specialized journal.”

To the contrary, we believe that a number of original concepts could appeal to a broad audience. For instance, the ideas of vesicles hitchhiking on transient, fast polymerizing actin filaments for local traffic, the interplay between local actin rails polymerization and actin-independent widespread mobility tuning, and the links between local resource reallocation and synaptic plasticity take us far beyond standard views of intracellular traffic.

Reviewer #2 Specific comments:

“2. No clear explanation for differential motility of FM- and QD- tagged vesicles is provided.”

We already provided a first answer to this question in the manuscript:

(L 196-198) “the modal velocity was ~20% greater for QD-labeled SVs than for FM-labeled SVs, but this makes sense if QD labeling allows the tracking of smaller, presumably more motile groups of vesicles than can be hardly detected with FM dye.”

This explanation is backed up by our measures of a significant negative correlation between cluster size and velocity (Fig S4E), which could match up with the recent finding that the traffic of larger cargos is physically hindered in thin axons (Wang et al., bioRxiv 492959, 2018). In the revised manuscript, we pushed forward this explanation by more explicitly referencing those correlation measures in the main body of the text [new L 184-185]. Importantly, if motion measures mildly differed at baseline, we are confident that the results obtained with the two labeling methods do no conflict but rather corroborate each other. Indeed, the response of FM-labeled SVs to nocodazole was paralleled by those of QD-labeled SVs (Figure 1).

“3. It remains uncertain are the labeled recycled vesicles represent SVs or some larger structures. QD labeling does not allow distinguishing between those.”

We concur with the Reviewer that providing extra concrete evidence and extensive bibliographical references to support the specificity of the labeling method towards recycling SVs would enhance the

clarity of our presentation. We have paid close attention to improving on this point— a key feature of our work – in the revised manuscript.

First, we now mention more clearly that our lab has extensive experience with specifically labeling recycling SVs with QDs (Zhang, Li and Tsien, Science 2009, Park, Li and Tsien, Science 2012) [L 171-174] and that we have previously performed an electron microscopy-based validation (Zhang, Cao and Tsien, PNAS 2007) (Rebuttal Fig 3) [new L 161].

Second, as stated above, our protocol is designed for improved selectivity towards recycling SVs (synaptotagmin 1 targeting and brief (90 s) neuronal stimulation, external quenching to remove any extracellular signal).

Third, we have performed new re-stimulation experiments that showed that SVs that contained a QD could exocytose again (Rebuttal Fig 2). We now include this extra validation in a Supplementary Figure of the revised manuscript (Fig S6) and further discuss the specificity of the labeling method [L 169-174].

“4. The authors state that activation of PKA by 10 μ M of forskolin (which is a rather hush treatment) resulted in elongation of the presynaptic cluster and that this matched the effect of synapsin 1 deletion. However, they do not confirm this in their own experiments.”

The Reviewer is right to note that this information is not directly provided in the manuscript. We however think it would be unfair and imbalanced to ask us to implement a laborious genetic protocol (Triple Knock Out Synapsin-deficient mice), to repeat a well-accepted result. Indeed, the disruption of synaptic vesicles clusters after synapsin deletion or phosphorylation has been consistently shown across many studies (Benfenati et al., Neuron 1992; Hosaka et al., Neuron 1999; Chi et al., Neuron 2003; Milovanovic et al., Science 2018) and our results (Fig 5A) are well-aligned with the literature. Moreover, we have shown that the application of okadaic acid, which regulates the phosphorylation of synapsin 1 at S9, identically to PKA activation (Huttner et al., J Biol Chem 1981; Czernik et al., PNAS 1987; Jovanovic et al., J Neurosci 2001), had an effect consistent with our interpretation (Fig 5 B-C).

“5. It has to be proven (556-558) that a reduction in actin polymerization affects presynaptic vesicle clustering. There is no evidence that supports this statement.”

We agree with the Reviewer that no direct evidence demonstrates this idea. But our intent was not to make this claim. Our original statement was:

“Our finding that vesicle motion depends on F-actin elongation provides a specific mechanism by which retrograde regulation of actin polymerization could in turn halt SV traffic and thus engender presynaptic vesicle clustering.” (L 556-558)

By which, we meant that the mechanisms we uncovered provided a fitting possible explanation for the presynaptic vesicle clustering observed by Wang and colleagues (Wang et al., Neuron 2005). We apologize for what appears to have been our lack of clarity and have now included a revised statement to re-explain our point [L 460-461].

Rebuttal Figure 1 (revised Fig S11): Sequences of Syt-IgG:QD active transport events are not directionally biased. (A) For 327 Syt1-IgG:QD clusters, we measured the net active transport- (AT)-based displacement as the distance between the QD positions before and after a sequence of AT events. Net displacement measures were then rescaled individually by normalizing each AT length by the average AT distance. Therefore, after normalization, the maximum net distance for a sequence of k AT events was k . (B) Net displacement values for different numbers of events were compared (red circles) to those obtained with an unbiased random-walk model (simulated trajectory $N=5000$) (left, average). AT distance values for the random walk model were randomly drawn in a centered normal distribution and renormalized for unit average norm. The anticipated maximal net displacement in case of a maximally directional sequence of AT events is a unit-slope line (dashed line). To better measure the evolution of the directional bias with an increasing number of AT events we also renormalized net displacement values by the maximal directional displacement (right). The directional bias for Syt1-IgG:QD measures did not appear to increase with the number of AT events and was indistinguishable from the unbiased random-walk model, in particular for long sequences of AT events which could more robustly highlight a possible directional bias. By contrast, the analysis of the AT of and 305 VAMP2:mCherry clusters provided evidence for strongly biased (~ 0.9) traffic (orange diamonds). (C) The distributions of experimental net displacement measures

were not significantly different from those of the unbiased random-walk for Syt1-IgG-containing clusters (6 AT events: rank-sum test $p>0.34$), but were for VAMP2:mCherry cargos 6 AT events: rank-sum test $p<10^{-6}$).

Rebuttal Figure 2 (revised Fig S6): Syt1-IgG:QDs are exocytosed upon electrical stimulation and the fluorescence of external QDs is quenched. (A) QD fluorescence quenching by BHQ3 was titrated in vitro using a plate fluorescence reader. (B) In neurons,

addition of 1 μ M BHQ3 to the bath resulted in the disappearance of multiple QDs, confirming the potent quenching of the fluorescence of external QDs. (C) When single electrical pulses were applied to cells by field stimulation, QD exocytosis appeared to be synchronized with the electrical stimulation. (D) When a train of electrical stimuli (1200 stimuli, 10 Hz) was applied a significant number of QDs disappeared, hence confirming the fusion-competence of SVs labeled with Syt1-IgG:QDs. Fluorescence was stable in the absence of stimulation. We selected immobile puncta for the analysis (N no stimulation = 133, N stimulation = 127).

Rebuttal Figure 3 (Modified from Zhang, Cao & Tsien, PNAS 2006): Electron microscopy confirming that QDs load in synaptic vesicles in a 1:1 ratio. “Only one Qdot is localized in the lumen of one synaptic vesicle. (A) Samples of Qdot-loaded synapse (+Qdots) and control (-Qdots). Red arrowheads point to vesicles containing electron dense puncta. (Scale bar, 500 nm.) (B) Distributions of luminal intensities (ROI indicated by dashed circles in Insets) of vesicles from four Qdot-loaded synapses (B1) and four control synapses (B2), with sum-of-Gaussian fits. A separate peak (red curve) is obvious in B1. (Insets) Exemplars of Qdot-positive vesicles and Qdot-negative vesicles at Qdot-loaded synapses and vesicles at control synapse. (Scale bar, 10 nm.)”

Rebuttal Figure 4 (revised Fig S5): FM dyes stain recycling synaptic vesicles and can be exocytosed by electrical and high K⁺ stimuli. (A) FM dyes assume a punctate distribution in cells when using a loading protocol based on high potassium stimulation. (B) A second high-potassium pulse triggered the exocytosis of FM dye molecules. Fluorescence in regions of interest automatically detected with ICY (red circles) strongly decreased after 90 s exposure to 90 mM K⁺ (median change -69.5%, $p < 10^{-6}$, sign test, N = 164). (C) The FM-dye distribution matched that of a fluorescent antibody against the luminal domain of synaptotagmin 1, which was preloaded in SVs. (D) A 10 Hz train of electrical stimuli caused massive FM dye exocytosis. The onset of the dye loss matched the start of the stimulus train (red bar), hence confirming the electrically-triggered release of SVs.

Rebuttal Figure 5 (modified from Harata et al., PNAS 2001): Electron microscopy imaging of synaptic vesicles stained with FM dyes. “Photoconversion of FM 1-43 in cultured hippocampal neurons [...] Representative image of a photoconverted bouton, previously subjected to field stimulation (20 Hz for 60 s). Dye exposure during stimulation and an additional 60 s of rest. Arrow points to a PC1 structure that was docked at the active zone.”

Rebuttal Figure 6 (revised Fig S16): Synaptic vesicle traffic is spared by microtubule disruption with colchicine. (A) Colchicine inhibited the traffic of neuronal mitochondria when we measured their total mobility (normalized distance traveled by directed motion), as expected because of their reliance on microtubules for traffic. Rank-sum test $p < 0.01$ (Veh. $N=11$, Colchicine $N=8$). (B) On the contrary, the mobility of SVs labeled with Syt1-IgG:QDs was spared by colchicine. Rank-sum test $p > 0.8$ (Veh. $N=8$, Colchicine $N=10$). (C) Unitary properties of Syt1-IgG:QDs active transport (AT) are compared. AT length $p > 0.1$. AT velocity: rank-sum test $p > 0.5$ (Veh. $N=721$, Colchicine $N=813$).

Rebuttal Figure 7 (revised Fig S17): SV traffic under myosin light chain kinase inhibition with ML-7. The traffic of SVs labeled with Syt1-IgG:QD was quantified in a paired experiment before and after treatment with 20 μM ML-7 for 15 min. (A) median mobility change was +7.0% ($p=1$, two-sided sign-test) and median AT frequency change was -9.0% ($p=1$, two-sided sign-test) ($N = 6$ coverslips). B: Unitary AT velocity was unchanged (median -0.007%, $p>0.87\%$, rank-sum test) while AT length saw a modest change (median -15%, $p<0.01$, rank-sum test) (N pre = 447; N post = 330).

Rebuttal Figure 8: Ciliobrevin D side-effects hamper its effective use as a dynein inhibitor. SVs labeled with Syt1-IgG:QD were imaged before and after a 5 min treatment with Ciliobrevin D. Ciliobrevin D IC_{50} is $\sim 15 \mu\text{M}$ (Firestone et al., Nature 2012). Two concentrations were tested: $20 \mu\text{M}$ (A) and $10 \mu\text{M}$ (B) and DMSO final concentration was $< 1/1000$. Both treatments consistently yielded almost complete loss of QD labels, pointing at important side-effects of Ciliobrevin D on synapse physiology even at concentration lower than IC_{50} .

REVIEWERS' COMMENTS:

Reviewer #1 (Remarks to the Author):

The comments and clarifications provided by the authors greatly helped reassuring the reviewer. Nevertheless, I fear that some additional work might still be needed as briefly explained below.

1- Preincubation for 2-4 hours with the anti-Syt antibody is still a point of contention as it suggests that the anti-Syt Ab needs to equilibrate along the endocytic pathway for an effect to be detected. My worry is that this in itself may affect the recycling of synaptic vesicles. as clustering of syt elicited by antibody binding could generate mistargeting in the endocytic pathway. Presumably, this step should not be needed to gather equivalent data. One suggestion would be to generate QD-streptavidin preincubated with the Ab, the excess Ab washed by centrifugation, and test whether the QD-biotin-streptavidin Ab promote similar effect on recycling vesicles.

2- The reviewer assumed that QDs were pH-sensitive from previous work on the subject by the authors' own laboratory (Zhang et al., Science, 2009). Quenchers are a good alternative and their use in the manuscript answered by queries.

3- The additional data on the FM1-43 destaining is also a point taken by the authors. I still consider the QD data more appropriate when it comes to tracking and welcome the suggestion of minimizing the FM work.

4- Another point taken related to the immobile fraction of the pools of synaptic vesicles as my initial worries stemmed from misunderstanding the staining protocol and the known delayed initiation of axonal retrograde trafficking following a short pulse of stimulation (Wang et al., J. Neuroscience 2014; Wang et al., Nat. Com. 2016). I was therefore expecting to see many more retrograde carriers.

Overall, the reviewer is positively impressed by the high quality of the rebuttal and by the additional work and clarification included in the revised version. I can only hope the author will value my last query (point 1) which, in my mind, is still a sticky, albeit minor point.

Reviewer #2 (Remarks to the Author):

The authors report on a finding that recycled synaptic vesicles may travel along axons using pulses of active transport utilizing an actin-dependent mechanism. They also report that this mechanism can be regulated by nitric oxide and protein kinase A. Those are interesting findings, which expands pilot observations that have been made earlier in invertebrate model systems. With additional experiments added to address the technical concerns of the referees, the original version of the paper is improved. The authors have honestly added previously published work to the reference list. The weakness of this work is in lack of the experiment(s) demonstrating the functional significance of their observations. How does it contribute to the synaptic function? Is it a developmental

phenomenon associated with synapse formation? Indeed, the authors provide nice hypotheses for potential roles in the discussion but did not put any effort to prove them. That is what readers expect from a paper from Nature Communications. I do not think the argument that this is very difficult to do is a good answer if one wants to publish at this level.

Reviewer #3 (Remarks to the Author):

The authors have addressed my concerns.

Tuesday, July 28, 2020

Reviewer #1 (Remarks to the Author):

The comments and clarifications provided by the authors greatly helped reassuring the reviewer. Nevertheless, I fear that some additional work might still be needed as briefly explained below.

1- Preincubation for 2-4 hours with the anti-Syt antibody is still a point of contention as it suggests that the anti-Syt Ab needs to equilibrate along the endocytic pathway for an effect to be detected. My worry is that this in itself may affect the recycling of synaptic vesicles. as clustering of syt elicited by antibody binding could generate mistargeting in the endocytic pathway. Presumably, this step should not be needed to gather equivalent data. One suggestion would be to generate QD-streptavidin preincubated with the Ab, the excess Ab washed by centrifugation, and test whether the QD-biotin-streptavidin Ab promote similar effect on recycling vesicles.

This is a reasonable suggestion, indeed for an approach that we had already adopted in a previous study (Park, Li & Tsien, *Science* 2012). We preincubated QD-streptavidin with biotinylated antibody against synaptotagmin in a dish, then applied to cells and drove uptake with electrical stimulation for 1 or 120 s. We observed punctuated vesicular transport over long distances using this approach, as exemplified in Figure A, below. While this staining protocol was suitable to study mobility and fusion at the single vesicle level (Park, Li & Tsien, *Science* 2012), it was less so in our hands for labeling a high fraction of the recycling pool of vesicles. In contrast, the two-step approach with a long exposure to Syt1-Abs, which we used in our new work, proved extremely efficient at labeling a large number of vesicles, which is required to

Figure A: Synaptic vesicles labeled with pre-ligated QD-Syt1-IgG probes display extra-synaptic long-range transport. Description from Park, Li & Tsien, *Science* 2012: "Single vesicles were efficiently labeled by use of streptavidin-coated Qdots conjugated to biotinylated antibodies against the luminal domain of the vesicular protein synaptotagmin 1. In an exemplar 3D trajectory (Fig. 1C), a single vesicle in a living neuron underwent ~12 s of intense movement, travelling almost unidirectionally with the net displacement of 3.2 mm over 90 s of imaging, with dwelling in two discrete zones, presumptive presynaptic terminals marked by distinct clouds of vesicles labeled with FM 4-64, a lipophilic probe for vesicular turnover"

obtain robust measures of axonal traffic as a majority of vesicles are confined to synaptic clusters rather than extrasynaptic.

Taking to heart the concern of the Reviewer about prolonged exposure to anti-Syt1 antibodies, we performed a comparison of the method used in the submitted work with a protocol with shortened application of the antibodies. The approach consisted of: (1) bathing cells in a Tyrode's solution with 25 mM K⁺ and containing synaptic AP5, NBQX, 4% BSA and the biotinylated **anti-Syt1 antibody for only 20 min**, (2) washing the cells for 10 min with an antibody-free 4 mM K⁺ solution, (3) applying a 45 mM K⁺ solution containing streptavidin-coated QDs and BSA for 90 s, (4) extensively washing with a QD- and Ab-free solution. Using mild K⁺ at step 1 was required because a short application of the antibody in the culture medium would yield insufficient spontaneous uptake. We measured the mobility of SVs stained with this protocol and compared it to the original approach with long exposure to the antibodies (4 coverslips each). **Quantification of the properties of active transport failed to show any significant difference between SVs exposed for a short and long time to the antibody** (mANOVA $p > 0.3$) (Figure B below). We have now included these results as Supplementary Fig. 6C in the revised manuscript to provide additional evidence of the validity of our experimental approach.

Finally, we would like to repeat a number of arguments that we have already made about the specificity of our staining protocol, despite the long incubation with Syt1-Abs. First, QDs are applied for only 90 s in a hyperkalemic solution to the cells pre-incubated with Syt1-Abs. If some antibodies were to end up in other compartments, their exocytosis during the short QD exposure would be unlikely, and therefore they would not be fluorescently labeled and confound our kinematic measures. Second, our mobility measures with QDs were in good agreement with that of compartments labeled with FM dyes (manuscript Figure 1), a widely-accepted marker of recycling synaptic vesicles. Third, we provided new evidence that QDs targeted structures that could exocytose upon rhythmic or single pulse electrical stimulation (manuscript Supplementary Figure 6), a hallmark of recycling synaptic vesicles.

2- The reviewer assumed that QDs were pH-sensitive from previous work on the subject by the authors' own laboratory (Zhang et al., Science, 2009). Quenchers are a good alternative and their use in the manuscript answered by [my] queries.

3- The additional data on the FM1-43 destaining is also a point taken by the authors. I still consider the QD data more appropriate when it comes to tracking and welcome the suggestion of minimizing the FM work.

We agree with the Reviewer that the quantum dot (QD)-based experiment has a number of important advantages over the FM-based one. We have therefore entirely removed FM-based results from main Figure 1. It has allowed us to put together a more compact and impactful figure with a direct comparison between VAMP2:mCherry and QD-based results. In the new version of the manuscript, we are proposing to move FM results in a supplementary figure of their own. Our motivation is two-fold: 1/ cross-check different experimental approaches (QD- and FM-based), which is fundamental to rigorous and reproducible science and 2/ provide a benchmark for the field as FM dyes are such a widely-accepted tool which were key to many seminal findings. We are convinced that this consensual presentation of results will convey our findings in the most convincing and rigorous manner. We thank the Reviewer for helping us improve the manuscript in this regard.

4- Another point taken related to the immobile fraction of the pools of synaptic vesicles as my initial worries stemmed from misunderstanding the staining protocol and the known delayed initiation of axonal retrograde trafficking following a short pulse of stimulation (Wang et al., J. Neuroscience 2014; Wang et al., Nat. Com. 2016). I was therefore expecting to see many more retrograde carriers.

We are glad this misunderstanding has been cleared up.

Overall, the reviewer is positively impressed by the high quality of the rebuttal and by the additional work and clarification included in the revised version. I can only hope the author will value my last query (point 1) which, in my mind, is still a sticky, albeit minor point.

Reviewer #2 (Remarks to the Author):

The authors report on a finding that recycled synaptic vesicles may travel along axons using pulses of active transport utilizing an actin-dependent mechanism. They also report that this mechanism can be regulated by nitric oxide and protein kinase A. Those are interesting findings, which expands pilot observations that have been made earlier in invertebrate model systems. With additional experiments added to address the technical concerns of the referees, the original version of the paper is improved. The authors have honestly added previously published work to the reference list. The weakness of this work is in lack of the experiment(s) demonstrating the functional significance of their observations. How does it contribute to the synaptic function? Is it a developmental phenomenon associated with synapse formation? Indeed, the authors provide nice hypotheses for potential roles in the discussion but did not put any effort to prove them. That is what readers expect from a paper from Nature Communications. I do not think the argument that this is very difficult to do is a good answer if one wants to publish at this level.

Although the editor has assured us that the newly shortened paper will not go back to the reviewers, we provide a brief reply just for the record. A large body of work shows that

presynaptic enhancement of neurotransmitter release and vesicle turnover accompanies NMDAR-dependent LTP in hippocampal cultures (references in main text of the MS). These neurons contain nitric oxide synthase postsynaptically for retrograde transmission (Wendland et al., PNAS 1994), and exhibit a large enhancement of activity-dependent uptake of markers of presynaptic vesicle turnover (Malgaroli et al., Science 1995) that is NMDAR dependent. Antonova, Hawkins, Wang, Kandel and colleagues showed that this presynaptic potentiation was associated in elevated levels of presynaptic vesicular proteins (for example, synaptophysin), apposed to postsynaptic sites undergoing increases in GluR1. Thus, postsynaptic and presynaptic aspects of NMDAR-dependent LTP were both anatomically and operationally aligned. Thus, functional effects attributable to postsynaptic activation, NO, and vesicle accumulation were already shown by work from Malgaroli, Tsien (Malgaroli et al., Science 1995) and colleagues by Hawkins, Antonova and coworkers (Antonova et al., Neuron 2001); the latter group also implicated cyclic GMP dependent kinase (downstream of NO), acting on Rho, ROCK and actin (Wang et al., Neuron 2005). Moreover, the question of presynaptic function modulation and SV re-distribution has recently come under the spotlight after Patzke, Südhof and colleagues showed that several neuromodulators, operating via PKA, alter the numbers of synaptic vesicles in hippocampal neurons from both mice and human (Patzke et al., Cell 2019). Thus, functional changes in vesicle abundance and participation in transmission have been recognized for several years (see Figure C for excerpts of few key findings), but what was lacking in all of this was *a mechanism to connect NO, cGMP and actin to vesicle mobilization*, precisely what our study provides.

Reviewer #3 (Remarks to the Author):

The authors have addressed my concerns.

We appreciate the reviewer's attention and straightforward response.